# LATENT-DPO: DIRECTION-AWARE PREFERENCE OPTIMIZATION FOR REASONING ALIGNMENT

## ABSTRACT

Although Large Language Models (LLMs) show impressive performance across diverse tasks, how to construct and effectively leverage high-quality supervision data remains an open challenge. While reverse question–answer pairs offer a means of data augmentation, models trained exclusively on forward–reverse mixtures through distillation still struggle to capture directional consistency. Standard Direct Preference Optimization (DPO) enforces uniform separation, often at the expense of shared reasoning structures. To address these limitations, we construct reverse examples and introduce Latent-DPO, an extension of preference optimization built upon reverse-augmented data. Latent-DPO incorporates a binary latent variable to model the consistency of reasoning paths and to modulate the DPO margin. This mechanism adaptively adjusts alignment strength, relaxing separation for pairs with subtle differences while maintaining a strong distinction for clearly divergent pairs. Empirical results demonstrate that our carefully constructed set of only 817 reverse examples produces a 4.5% average improvement across five benchmarks. Moreover, Latent-DPO yields consistent improvements across multiple datasets and base models, achieving average accuracy gains of up to 3.2%. Our code and data are available at the anonymous repository: https://anonymous.4open.science/r/submission_429.

## 1 INTRODUCTION

Large Language Models (LLMs) have demonstrated impressive capabilities across diverse language tasks (Brown et al., 2020; Touvron et al., 2023; OpenAI, 2023), but most alignment efforts emphasize *forward reasoning*. In this setting, conclusions are derived step by step from premises—for example, computing the area of a rectangle from its base and height. In contrast, *reverse reasoning*, such as inferring a missing base from the area and height, has received far less attention. Human cognition is inherently bidirectional: when solving problems, planning, or proving theorems, people naturally combine forward deduction with backward inference (Newell et al., 1972; Al-Ajlan, 2015; Jara-Ettinger, 2019). This underscores the need to integrate reverse reasoning as a complementary element for more robust and generalizable alignment.

Building on this foundation, a growing body of work has incorporated reverse supervision to enhance LLM reasoning across diverse domains. For reverse data generation, MathGenie (Lu et al., 2024) applies solution-to-question back-translation to augmented solutions, while Reverse Thinking (Chen et al., 2024) shows that training with paired forward and backward exemplars improves commonsense and logical reasoning. In optimization, OptiBench (Yang et al., 2024) spans linear and nonlinear tasks, with its ReSocratic method boosting performance by back-translating demonstrations. For verification and curriculum learning, FOBAR (Jiang et al., 2024) generates backward verification questions by masking parts of problems; $R^3$ (Xi et al., 2024) applies reverse curriculum reinforcement learning, sliding start states from demonstration ends to the beginning for finer-grained outcome supervision; and RCOT (Xue et al., 2023) reconstructs problem statements from solutions to detect and correct inconsistencies. Broader reverse paradigms include Reason-from-Future (Xu et al., 2025), which alternates forward and backward reasoning to improve long-chain math accuracy; reconstructing questions from answers to improve causal hypothesis tasks (Ranaldi et al., 2025); and reverse exemplar generation, shown to improve few-shot prompting and geometry reasoning (Wang et al., 2025; Deng et al., 2024; Deb et al., 2024).

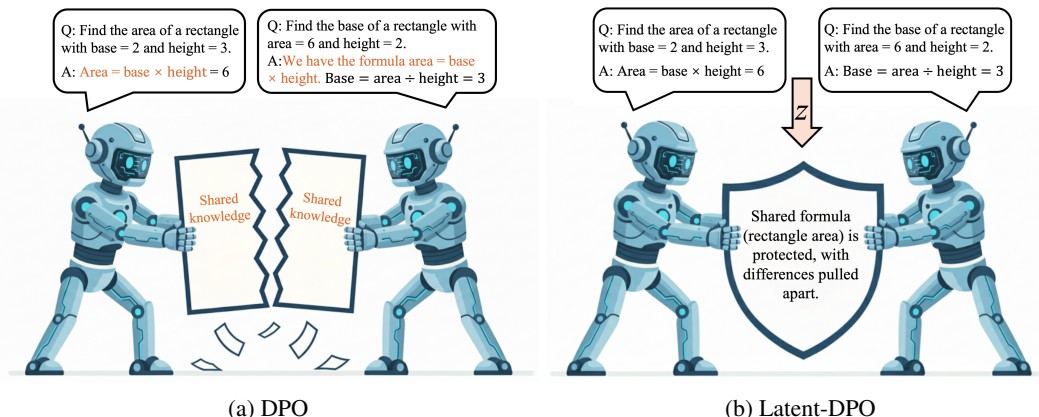

(a) DPO                   (b) Latent-DPO

Figure 1: Comparison between DPO and Latent-DPO. (a) DPO pulls apart both shared and different parts of reasoning. (b) Latent-DPO preserves shared knowledge while capturing directional differences. The latent variable $z$ models the directional consistency of reasoning paths, adaptively adjusting the strength of alignment between consistent and divergent responses.

While these reverse-supervision approaches demonstrate notable improvements, they typically rely on powerful teacher models to generate exemplars and offer little guidance on how LLMs themselves should distinguish between forward and backward chains of thought. This challenge is further compounded by the *Reversal Curse* (Berglund et al., 2023), which shows that transformers often fail to generalize under task inversion due to inadequate entity binding, even when exposed to substantial data. Architectural remedies such as the Joint-Embedding Predictive Architecture (Wang & Sun, 2025) introduce additional memory mechanisms to alleviate this issue, but their strong inductive biases and heavy modifications limit their applicability. A more lightweight alternative is to apply Direct Preference Optimization (DPO) to enforce separation between reasoning directions. However, DPO treats all differences indiscriminately and inadvertently erodes shared knowledge. As illustrated in Figure 1, a rectangle-area problem can be posed in two directions: in the forward form, the task is to compute the area from the base and height, and in the backward form, the task is to recover the base from the area and height. Both directions rely on the same rectangle-area formula. Standard DPO pushes apart not only direction-specific operations (multiplication to compute area vs. division to solve for the base) but also the rectangle-area formula, thereby undermining the shared knowledge across them.

To tackle the above challenges, we expand high-quality datasets through reverse data construction and extend preference optimization with a latent-variable formulation that explicitly models whether a response follows the intended reasoning path. A binary latent variable $z$ captures the consistency of reasoning paths and is incorporated into the DPO margin. As illustrated in Figure 1(b), this design enables Latent-DPO to preserve the rectangle-area formula shared across both directions, while maintaining separation between direction-specific operations (multiplication for area vs. division for the base). In practice, pairs with subtle differences receive milder separation, whereas pairs with clear divergence are pushed further apart. This adaptive modulation mitigates the over-penalization of partially aligned responses, allowing the model to retain transferable reasoning structures while enforcing direction-aware alignment. We summarize our contributions as follows.

- **Reverse Data Augmentation**: from 817 curated problems in the LIMO dataset (Ye et al., 2025), we construct three distinct reverse subsets of 817 questions each. Distillation on each subset achieves 3.1% to 4.3% accuracy improvements over the base model.

- **Latent Direction Alignment**: we extend DPO by incorporating a latent variable $z$ that models reasoning-direction consistency, thereby facilitating the disentanglement of direction-specific alignment signals while maintaining shared knowledge.

- **Empirical Evaluation**: we evaluate Latent-DPO across multiple base models and benchmarks, observing consistent improvements. Average accuracy gains range from 0.9% to 3.2% across different datasets and model foundations. We further conduct ablation studies and analyze the impact of reverse-data scale on alignment performance.

## 2 RELATED WORK

Our discussion focuses on two closely related lines of research: reverse supervision and preference optimization. Additional background on data quality, efficient scaling, and inference-time diversification is provided in Appendix A.

**Reverse Supervision.** Recent work explores reverse reasoning alongside forward supervision: MathGenie (Lu et al., 2024) performs back-translation from solutions to questions, RevThink (Chen et al., 2024) distills paired forward–backward exemplars, and verification frameworks such as FO-BAR (Jiang et al., 2024) and RCoT (Xue et al., 2023) employ reverse formulations for self-checking and consistency, while $R^3$ (Xi et al., 2024) uses a reverse curriculum in reinforcement learning to provide stepwise supervision from outcome signals. The OptiBench (Yang et al., 2024) benchmark and its ReSocratic data synthesis method generate problems by reversing from solutions. Other directions include backward reasoning toward causal hypotheses (Ranaldi et al., 2025), reverse-style abductive inference (Deb et al., 2024), and self-generated reverse exemplars for few-shot prompting and geometry reasoning (Wang et al., 2025; Deng et al., 2024). While these studies highlight the potential of reverse data, their application to LLM alignment has been insufficiently explored.

**Direct Preference Optimization.** Direct Preference Optimization (DPO) (Rafailov et al., 2023) frames preference alignment as a supervised Bradley–Terry modeling problem, avoiding explicit reward models and unstable policy gradients. Variants extend DPO along multiple axes: reference-free (KTO (Ethayarajh et al., 2024), SimPO (Meng et al., 2024)); richer supervision via step-level margins (Lai et al., 2024), process feedback (Li et al., 2024; Lightman et al., 2023), or trajectory-level optimization (Jiao et al., 2024); and stability improvements through Multi-Objective-DPO (Zhou et al., 2023). Robust Preference Optimization through Reward Model Distillation (Fisch et al., 2024) uses distillation from reward models and ensemble uncertainty to enhance robustness to annotation bias and mitigate policy degeneration. While Reinforcement Learning from Human Feedback (RLHF) and Reinforcement Learning from AI Feedback (RLAIF) have driven recent progress in alignment (Bai et al., 2022a; Stiennon et al., 2020; Ouyang et al., 2022), most DPO variants treat winning and losing responses as entirely disjoint, neglecting their shared reasoning structures. This, combined with evidence that transformers fail to generalize under task inversion (Wang & Sun, 2025), underscores the need for direction-consistency modeling and motivates our Latent-DPO approach to capture shared reasoning signals while adaptively aligning bidirectional reasoning.

## 3 METHODOLOGY

In this section, we outline the pipeline for constructing a bidirectionally paired dataset (Section 3.1); afterward, we present a latent variable framework to model reasoning-path alignment (Section 3.2); finally, we derive the Latent-DPO objective with KL regularization for robust training (Section 3.3).

### 3.1 REVERSE DATA CONSTRUCTION

We begin with the 817 curated problems in the LIMO dataset (Ye et al., 2025), whose small scale and careful curation make it well suited for investigating reasoning alignment without requiring large-scale data construction. Building on this foundation, we generate reverse-style counterparts through a multi-stage pipeline of generation and validation, as illustrated in Figure 2. For each original examples, DeepSeek V3 (Liu et al., 2024) generates three distinct reverse questions, allocated to three subsets. Thus, the 817 LIMO problems produce three reverse subsets, each with 817 questions. Each reverse question is then solved by Qwen3-32B (Yang et al., 2025), producing complete reasoning traces and final answers. Because the first subset is used both for knowledge distillation and Latent-DPO training, it undergoes an additional filtering step with Qwen3-7B (Yang et al., 2025): whenever an answer is judged incorrect, the corresponding reverse question is resubmitted to Qwen3-32B until a correct solution is obtained. In contrast, the other two subsets are not primarily used for distillation and are retained in their original outputs from Qwen3-32B, preserving the teacher's raw generations and their inherent stylistic characteristics in Latent-DPO training. The full prompt templates and data collection configurations are provided in Appendix C.

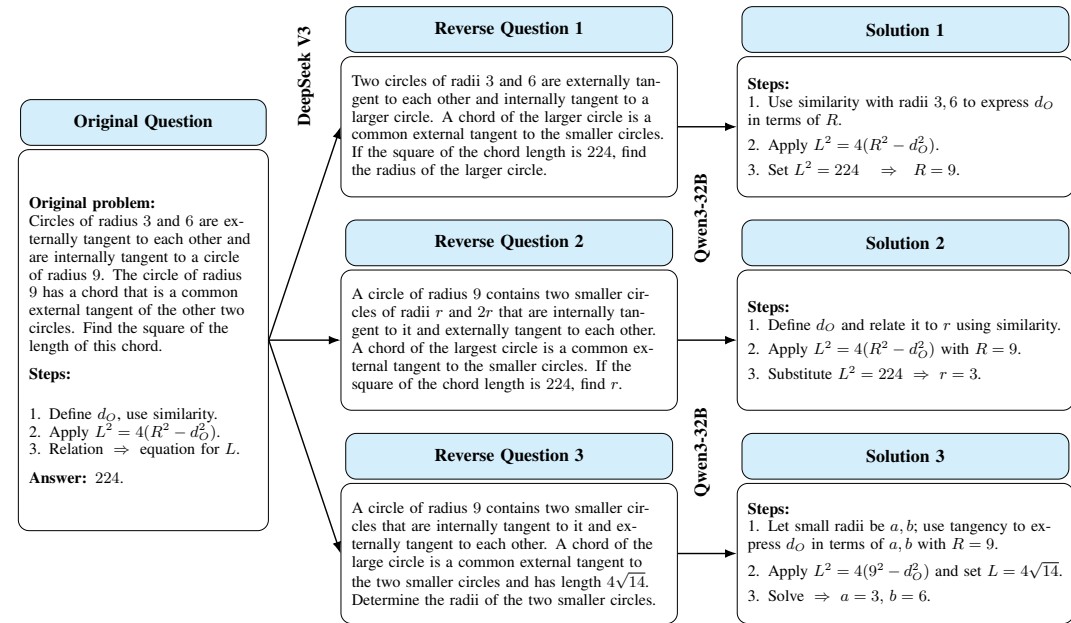

Figure 2: Pipeline for reverse data construction: DeepSeek V3 generates reverse questions, and Qwen3-32B provides corresponding solutions.

In our constructed dataset for Latent-DPO, every prompt $x$ (whether forward or reverse) is paired with both a direction-consistent response and a direction-divergent response, forming a tuple $(x, y^+, y^-)$. For a forward problem, the original solution serves as $y^+$, while the reverse-generated solution provides $y^-$. For a reverse problem, the reverse solution produced by Qwen3-32B is treated as $y^+$, whereas the original forward solution (together with other mismatched responses) serves as $y^-$. This bidirectional pairing encodes both alignment and divergence, supplying the supervision required for direction-aware preference optimization.

## 3.2 LATENT VARIABLE FOR REASONING-PATH ALIGNMENT

**Latent Alignment Modeling.** In structured reasoning tasks (e.g., mathematics and logic), dis-preferred responses may apply the correct principle but fail to address the prompt. Treating such partially relevant cases as fully incorrect risks impairing the model's ability to retain shared reasoning knowledge across related problems. To address this challenge, we introduce a binary latent variable $z \in \{0, 1\}$, where $z = 1$ denotes that the response $y$ is directionally consistent with the prompt $x$, and $z = 0$ indicates a deviation from the intended reasoning trajectory.

To infer $z$, we concatenate the prompt and the initial segment of its response, denoted by $(x; y)$. The resulting sequence is truncated to a fixed length, and the hidden state of the last non-padding token is extracted from the policy model $\pi_\theta$, yielding the representation $h_\theta(x, y)$. Truncation is motivated by the intuition that early tokens typically reveal reasoning alignment with the prompt, while later ones often introduce noise. The representation $h_\theta(x, y)$ is then passed to a shared posterior head parameterized by $\phi$, which outputs a binary distribution:

$$q_\phi(z \mid x, y) = \text{softmax}\big(u_\phi(h_\theta(x, y))\big), \quad u_\phi(\cdot) \in \mathbb{R}^2,$$

where $q_\phi(z = 1 \mid x, y)$ is the posterior probability that the response is consistent with the intended reasoning path.

**Remark 1.** *The posterior parameters $\phi$ are optimized directly through the overall loss, while the policy parameters $\theta$ are updated solely via the preference surrogate (introduced in Section 3.3). To decouple the two objectives, gradients from the posterior head are stopped at $h_\theta(x, y)$, preventing them from propagating into the policy encoder.*

Table 1: Key notation for latent-variable DPO, with a rectangle-area example (Q&A in Figure 1).

| Symbol | Description | Example |
|---|---|---|
| $x$ | Input prompt (problem) | Find the base of a rectangle |
| $y^+$ | Correct, direction-matched response | Base = area ÷ height |
| $y^-$ | Incorrect or off-task response | Area = base × height |
| $z = 1$ | Latent variable indicating path-aligned response | $x \to y^+$ |
| $z = 0$ | Latent variable indicating off-topic response | $x \to y^-$ |
| $q_w$ | Variational posterior probability that $y^+$ is path-aligned | High $\to$ consistency |
| $q_l$ | Variational posterior probability that $y^-$ is path-aligned | Low $\to$ divergence |

**Factorized Adaptive Latent Modulation.** Now we define $z \triangleq (z_w, z_l)$ and model the posterior over preferred and dispreferred responses using a factorized form:

$$q_\phi(z_w, z_l \mid x, y^+, y^-) \;=\; q_\phi(z_w \mid x, y^+)\, q_\phi(z_l \mid x, y^-), \tag{1}$$

where $z_w$ and $z_l$ indicate the alignment of the preferred ($y^+$) and dispreferred ($y^-$) responses, respectively. Both responses share the same classifier head (details in Appendix B.5), ensuring evaluation in a consistent feature space without explicitly modeling dependencies.

For notational convenience, we define

$$q_w \triangleq q_\phi(z{=}1 \mid x, y^+) \;\; \text{and} \;\; q_l \triangleq q_\phi(z{=}1 \mid x, y^-), \tag{2}$$

so that $q_w$ and $q_l$ denote the posterior probability that the preferred and dispreferred responses are aligned, respectively. Intuitively, the latent posteriors act as modulation factors on the preference signal: when the estimates are confident (e.g., $q_w \to 1$, $q_l \to 0$), the loss collapses to standard DPO, whereas in ambiguous cases the modulation adaptively relaxes the margin, preventing undue penalization of shared reasoning knowledge.

To ensure that this modulation mechanism functions effectively and does not collapse prematurely, we regularize $q_\phi$ toward a uniform Bernoulli prior $\texttt{Unif}(\{0,1\})$ via a KL term, i.e., $\text{KL}\big[q_\phi(\cdot) \,\|\, \texttt{Unif}(\{0,1\})\big]$. This regularization directly biases $q_\phi$ toward uniformity, thereby promoting exploration and preventing premature overconfidence of the posterior during training.

### 3.3 LATENT-DPO

**Deriving the objective of Latent-DPO.** To explicitly model the reasoning-path consistency of preference alignment, we introduce the latent variables $z = (z_w, z_l)$ defined in Section 3.2, and accordingly define the corresponding DPO margin as:

$$m(z) = z_w\big[\ell_\theta(y^+ \mid x) - \ell_\text{ref}(y^+ \mid x)\big] - (1 - z_l)\big[\ell_\theta(y^- \mid x) - \ell_\text{ref}(y^- \mid x)\big], \tag{3}$$

where $\ell_\theta(\cdot \mid x)$ and $\ell_\text{ref}(\cdot \mid x)$ denote the unnormalized sums of token log-likelihoods over response tokens under the trainable policy $\pi_\theta$ and the reference model $\pi_\text{ref}$, respectively.

We summarize the advantages of introducing latent variables in DPO as follows:

- When $z_w = 1$, $y^+$ is considered reasoning-aligned and contributes positively to the margin; otherwise, $y^+$ is treated as misaligned and its contribution is excluded from this margin.
- When $z_l = 0$, $y^-$ is considered reasoning-misaligned, thus contributing negatively to the margin; otherwise, $y^-$ is treated as aligned, and it should not be penalized in the margin.
- When $z_w = 1$ and $z_l = 0$, the Equation 3 reduces to the standard DPO objective, thereby preserving consistency with the original method.

Given the unobservability of $z$, we define the posterior-modulated margin as the expectation of $m(z)$ under the posterior distribution $q_\phi$ (Equation 1), which is formally given by:

$$m_\text{eff} = \mathbb{E}_{q_\phi}\big[m(z)\big] = q_w\big[\ell_\theta(y^+ \mid x) - \ell_\text{ref}(y^+ \mid x)\big] - (1 - q_l)\big[\ell_\theta(y^- \mid x) - \ell_\text{ref}(y^- \mid x)\big]. \tag{4}$$

Finally, we optimize the DPO surrogate loss defined on the posterior-modulated margin in Equation 4 and complement it with a Uniform-KL that acts directly on both alignment posteriors:

$$\mathcal{L}_\text{Latent-DPO} = \mathbb{E}_{(x,y^+,y^-) \sim \mathcal{D}}\Big[-\log \sigma\big(\beta\, m_\text{eff}\big) + \lambda_\text{kl} \cdot \text{KL}\Big[q_\phi(\cdot \mid x, y^+, y^-) \,\|\, \texttt{Unif}(\{0,1\})^2\Big]\Big], \tag{5}$$

where $\sigma(t) = 1/(1 + e^{-t})$ is the sigmoid function, $\beta$ is an inverse temperature controlling the sharpness of the loss and $\lambda_{\text{kl}}$ controls the regularization strength. The procedure is given in Algorithm 1.

---

**Algorithm 1** Latent-DPO Training

---

**Input:** Training set $\{(x, y^+, y^-)^{(i)}\}_{i=1}^N$; policy model $\pi_\theta$; reference model $\pi_{\text{ref}}$; learning rate $\eta$
  1: **Initialize** $\theta, \phi$
  2: **for** each minibatch $(x, y^+, y^-)$ **do**
  3:     Infer the alignment posteriors $q_w$ and $q_l$ by Equation 2
  4:     Compute the log-likelihoods under $\pi_\theta$ and $\pi_{\text{ref}}$ for $y^+$ and $y^-$
  5:     Form the posterior-modulated margin $m_{\text{eff}}$ by Equation 4
  6:     Compute the total loss with Uniform-KL by Equation 5
  7:     Update $\theta \leftarrow \theta - \eta \nabla_\theta \mathcal{L}$ and $\phi \leftarrow \phi - \eta \nabla_\phi \mathcal{L}$          ▷ stop posterior gradient for $\theta$
  8: **end for**
  9: **return** Policy model $\pi_\theta$

---

**Theoretical Properties of Latent-DPO.** We now analyze the theoretical properties of Latent-DPO, focusing on its variational interpretation and approximation quality. For convenience, let $f(m) = -\log \sigma(\beta m)$ denote the logistic loss on margin $m$, which is a convex function on $\mathbb{R}$.

**Proposition 1.** *The logistic surrogate objective $-\log \sigma(\beta m_{\text{eff}})$ is a variational lower bound of the strict variational objective $\mathbb{E}_{q_\phi}[f(m(z))]$.*

*Proof.* Since $f(\cdot)$ is convex on $\mathbb{R}$ and $q_\phi(z)$ is a valid probability distribution over $z$, we have

$$\underbrace{\mathbb{E}_{q_\phi}[f(m)] \geq f(\mathbb{E}_{q_\phi}[m])}_{\text{Jensen's inequality}} \overset{(4)}{=} f(m_{\text{eff}}) = -\log \sigma(\beta \, m_{\text{eff}}).$$

Hence, $-\log \sigma(\beta m_{\text{eff}})$ forms a meaningful, tractable surrogate objective for optimization. □

**Proposition 2.** *The Jensen gap between the strict objective and its surrogate is bounded by the posterior variance of the margin: $\mathbb{E}_{q_\phi}[f(m)] - f(\mathbb{E}_{q_\phi}[m]) \leq \frac{\beta^2}{8} \text{Var}_{q_\phi}[m(z)]$.*

*Proof Sketch.* By Taylor's theorem, there exists some $\xi$ between $m(z)$ and $m_{\text{eff}}$ such that

$$f(m(z)) = f(m_{\text{eff}}) + f'(m_{\text{eff}})(m(z) - m_{\text{eff}}) + \frac{1}{2} f''(\xi)(m(z) - m_{\text{eff}})^2. \tag{6}$$

Taking expectations of Equation 6 and using $f''(t) = \beta^2 \sigma(\beta t)\sigma(-\beta t) \leq \beta^2/4$ gives the result. □

**Remark 2.** *The logistic surrogate can be formally derived as a variational Evidence Lower Bound (ELBO; Blei et al. 2017) under a latent-variable extension of the Bradley–Terry preference model (Bradley & Terry, 1952); see Appendix B.1 for the full derivation.*

# 4 EXPERIMENTS

## 4.1 EXPERIMENTAL SETUP

**Training Configuration.** All experiments are implemented using the SWIFT framework (Zhao et al., 2024). Supervised fine-tuning (SFT) is conducted for 3 epochs with a learning rate of $1 \times 10^{-5}$ and a maximum sequence length of 11,000 tokens. Latent-DPO training is performed for 3 epochs with a lower learning rate of $1 \times 10^{-6}$ and a sequence length capped at 1,000 tokens. All models are trained in **bfloat16** precision on $4 \times$RTX 4090 (24GB) GPUs.

**Evaluation.** We select five benchmarks to comprehensively evaluate reasoning alignment across both distributional alignment and generalization. OpenAI Math 500 (Hendrycks et al., 2021a; Lightman et al., 2023) serves as an *in-domain* benchmark, closely reflecting the algebraic, geometric, and symbolic reasoning style of LIMO. We regard AIME-25 (Committee, 2025) as *near-domain*, sharing LIMO's competition style but differing in distribution and yearly formats. To further evaluate generalization, we include three *out-of-domain* benchmarks: GPQA (Rein et al., 2024), which probes

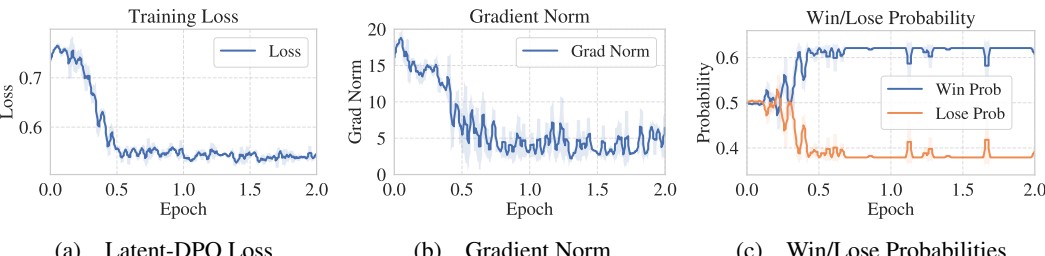

| (a)   Latent-DPO Loss | (b)   Gradient Norm | (c)   Win/Lose Probabilities |

Figure 3: Training dynamics of Latent-DPO. Panel (a) tracks the optimization loss across epochs, (b) illustrates gradient norm stability, and (c) shows win/lose probabilities, obtained as $\sigma(\beta \cdot m_{\text{eff}})$ where the $m_{\text{eff}}$ is the difference between sequence-level log-likelihoods of $y^+$ and $y^-$. This is equivalent to a two-class softmax over teacher-forced scores. Shaded regions indicate variability across runs.

graduate-level science and conceptual reasoning; Gaokao MathQA (GMQ) (Zhang et al., 2023), consisting of high-school mathematics word problems; and Leaderboard Math: Math Geometry Hard (LMGH) (Hendrycks et al., 2021b), a collection of advanced geometry problems. Together, these benchmarks span multiple domains, levels, and problem formats, offering a comprehensive assessment of reasoning alignment.

We evaluate performance using the pass@1 accuracy metric in a zero-shot chain-of-thought setting. Decoding is performed with greedy sampling, with maximum output lengths set to 16,000 tokens for AIME-25 and 10,000 tokens for the other benchmarks. All evaluations follow the standard lm-eval-harness implementation (Gao et al., 2024), which applies rule-based matching through canonicalized string normalization and numerical equivalence. This unified evaluation protocol ensures objective and fully reproducible comparisons across diverse reasoning benchmarks. Additional training and evaluation details are provided in Appendix D.

## 4.2 Effectiveness of Reverse Data and Latent DPO

Before evaluating Latent-DPO, we first assess the quality of the constructed reverse data through SFT experiments on Qwen3-1.7B-Base and Qwen3-1.7B, the latter further optimized with Reinforcement Learning (RL). We distill models on each reverse subset individually and analyze the effect of mixing forward (LIMO) and reverse data at different ratios. We also compute the verified accuracy of each subset using Qwen3-7B as a judge, with detailed statistics and ablations reported in Appendix F. As shown in Table 2, and further supported by the analyses in Tables 5 and 7, reverse data matches the quality of original LIMO data in distillation, underscoring its value for reasoning alignment. However, SFT trained on mixed forward and reverse data performs worse than single-direction SFT, indicating that direct combination can introduce interference rather than synergy. Moreover, Table 6 shows that distillation on the RL-trained Qwen3-1.7B model consistently degrades performance across all reverse subsets, underscoring a conflict between reverse-data distillation and prior RL-alignment. Collectively, these findings reveal the need for alignment strategies that effectively leverage high-quality reverse data.

We conduct Latent-DPO experiments on three model foundations: Qwen3-1.7B-Base, the official Qwen3-1.7B model, and an SFT model trained on a mixture of the original LIMO data and reverse-filtered augmentations. Importantly, since the official Qwen3-1.7B has already undergone RL and extensive alignment, further gains on this strong baseline underscore the complementary strengths of Latent-DPO. For the initial configuration of our experiments, Latent-DPO is trained on 1,634 preference pairs, constructed by pairing each of the 817 LIMO problems with its corresponding reverse counterpart. For each original LIMO problem, the ground-truth forward solution is treated as the preferred response ($y^+$), while the corresponding reverse solution is dispreferred ($y^-$). Conversely, for each generated reverse problem, the reverse solution is designated as $y^+$ and the forward solution as $y^-$. Figure 3 summarizes the training dynamics of Latent-DPO in detail. Panel (a) depicts the smooth convergence of the optimization loss during training. Panel (b) illustrates the consistently stable behavior of gradient norms throughout the entire process, while Panel (c) highlights the progressively widening gap between direction-consistent and direction-divergent responses.

Table 2: Performance comparison of Latent-DPO across three model families: Qwen3-1.7B-Base, SFT on mixed forward and reverse data followed by Latent-DPO, and the official Qwen3-1.7B release. This table also compares SFT configurations with varying datasets. Within the Qwen3-1.7B-Base family, the best result is highlighted in **bold**, and the second-best is marked with underlining.

| Base Model | Training Strategy | AIME-25 | GPQA | Math 500 | GMQ | LMGH | Avg. Acc. |
|---|---|---|---|---|---|---|---|
| Qwen3-1.7B-Base | None | 0% | 28.3% | 45.8% | 33.6% | 2.3% | 22% |
| Qwen3-1.7B-Base | SFT (LIMO) | 0% | 27.8% | 47.2% | 33.6% | **13.6%** | 24.5% |
| Qwen3-1.7B-Base | SFT (Reverse) | 3.3% | **30.3%** | 46.0% | 33.6% | 12.1% | 25.1% |
| Qwen3-1.7B-Base | SFT (Mixed) | 0% | **30.3%** | 47.4% | 33.1% | 7.6% | 23.7% |
| SFT (Mixed) | Vanilla DPO | 0% | 28.8% | 45.2% | 32.8% | 8.3% | 23.0% |
| SFT (Mixed) | Latent-DPO (Ours) | **6.7%** | 29.8% | **48.6%** | **35.3%** | 12.1% | **26.5%** |
| Qwen3-1.7B-Base | None | 0% | 28.3% | 45.8% | 33.6% | 2.3% | 22% |
| Qwen3-1.7B-Base | Latent-DPO (Ours) | **6.7%** | 27.8% | 46.8% | 34.8% | 3.8% | 24.0% |
| Qwen3-1.7B | None | 13.3% | 29.8% | 47.6% | 35.9% | 10.6% | 27.4% |
| Qwen3-1.7B | Latent-DPO (Ours) | 16.7% | 28.3% | 49.2% | 35.9% | 13.6% | 28.7% |

The main experimental findings are summarized in Table 2. For the Qwen3-1.7B-Base model, Latent-DPO improves average accuracy from 22% to 24%, corresponding to a 2% gain. When applied to the SFT-trained model on mixed data, Latent-DPO further raises performance to 26.5%, yielding a 2.8% improvement over SFT. On the official Qwen3-1.7B model, Latent-DPO still increases accuracy from 27.4% to 28.7%. These results demonstrate that Latent-DPO provides consistent and reliable improvements under a limited supervision budget of 1,634 examples, with gains driven by contrastive alignment that enforces direction-aware separation between forward and reverse reasoning. To further validate robustness and reliability of our results, we repeat each Latent-DPO experiment three times with different random seeds and report mean accuracy and standard deviation across runs in Table 8.

Table 3 reports ablations on the critical components of Latent-DPO. The term *Uniform-KL* refers to a KL divergence between the learned posterior and a uniform Bernoulli prior, introduced to mitigate posterior collapse. Our base experimental configuration incorporates both Uniform-KL regularization and a differentiable posterior. Removing either component leads to clear performance degradation across benchmarks. In particular, removing the Uniform-KL term results in a modest performance decline of 1.1%-2%, whereas removing the posterior from gradient flow causes a more significant degradation, ranging from 3.1% to 4.7%. Overall, the ablation evidence confirms that Uniform-KL regularization and the differentiable posterior are critical components, providing the foundation for Latent-DPO's consistent and robust improvements over vanilla DPO.

Table 3: Ablation study on different training settings. **Ours** setting uses Uniform-KL regularization and a differentiable posterior. We ablate by removing the Uniform-KL regularization (w/o Uniform-KL) and by removing the differentiable posterior (w/o Posterior). The percentages with ↓ indicate the average performance drop relative to **Ours**. Within each base model family, best results in **bold**.

| Base Model | Setting | AIME-25 | GPQA | Math 500 | GMQ | LMGH | Avg. Acc. |
|---|---|---|---|---|---|---|---|
| Qwen3-1.7B-Base | Ours | **6.7%** | **27.8%** | **46.8%** | 34.8% | **3.8%** | **24.0%** |
| | w/o Uniform-KL | 3.3% | 26.8% | 45.8% | **36.5%** | 1.5% | 22.9%(1.1%↓) |
| | w/o Posterior | 0% | 26.3% | 42.2% | 33.6% | 2.3% | 20.9%(3.1%↓) |
| Qwen3-1.7B | Ours | **16.7%** | **28.3%** | **49.2%** | **35.9%** | **13.6%** | **28.7%** |
| | w/o Uniform-KL | 13.3% | 25.8% | 47.4% | 35.6% | 11.4% | 26.7%(2.0%↓) |
| | w/o Posterior | 10.0% | 26.8% | 44.6% | 35.6% | 3.0% | 24.0%(4.7%↓) |

## 4.3 THE IMPACT OF THE NUMBER OF REVERSE PROBLEMS ON MODEL PERFORMANCE

Building on the reverse subsets introduced in Section 3.1, Table 5 shows that distilling on any individual reverse subset achieves improvements of 3.1% to 4.5% over the base model. These results confirm that reverse examples provide high-quality supervision and establish a solid foundation for assessing the effectiveness of Latent-DPO under multi-subset augmentation. In contrast, Table 6 fur-

Table 4: Impact of the number of reverse problems per forward exemplar on model performance. Here, Number indicates the problem set size: 0 refers to the officially released pretrained model without additional training, 2 corresponds to the full LIMO forward set augmented with one corresponding reverse set, 3 augmented with two reverse sets, and 4 with three reverse sets. Within each base model family, the best result is shown in **bold** and the second-best is underlined. An upward arrow (↑) indicates the improvement in average accuracy relative to the base model.

| Base Model | Number | AIME-25 | GPQA | Math 500 | GMQ | LMGH | Avg. Acc. |
|---|---|---|---|---|---|---|---|
| Qwen3-1.7B-Base | 0 | 0% | 28.3% | 45.8% | 33.6% | 2.3% | 22% |
| | 2 | 6.7% | 27.8% | 46.8% | 34.8% | **3.8%** | 24.0%(2.0%↑) |
| | 3 | **10.0%** | 28.8% | **48.8%** | 35.3% | 3.0% | **25.2%**(3.2%↑) |
| | 4 | 0% | 28.8% | 48.2% | **32.7%** | 4.6% | 22.9%(0.9%↑) |
| Qwen3-1.7B | 0 | 13.3% | **29.8%** | 47.6% | 35.9% | 10.6% | 27.4% |
| | 2 | 16.7% | 28.3% | 49.2% | 35.9% | **13.6%** | 28.7%(1.3%↑) |
| | 3 | 16.7% | 27.8% | 49.0% | 35.9% | 12.9% | 28.4%(1.0%↑) |
| | 4 | **20.0%** | 28.3% | **51.4%** | **36.2%** | 9.9% | **29.2%**(1.8%↑) |

ther reveals that applying distillation to models already optimized with RL-based preference training leads to substantial performance degradation. Moreover, Table 7 demonstrates that directly mixing forward and reverse data during SFT leads to consistent regressions, suggesting that distillation on combined data is not an effective alignment strategy. Accordingly, our subsequent experiments focus on Latent-DPO under varying augmentation configurations, where the original forward set is systematically augmented with different numbers of reverse subsets.

Table 4 reports the impact of augmenting the LIMO forward dataset with different numbers of reverse subsets on Qwen3-1.7B-Base and Qwen3-1.7B under Latent-DPO training. For Qwen3-1.7B-Base, reverse augmentation yields clear but non-monotonic gains: accuracy rises from 22% to 24% with two subsets and peaks at 25.2% with three, but drops to 22.9% with four subsets. This pattern indicates that the base model, constrained by limited capacity, benefits from moderate directional diversity. However, when the augmentation becomes excessive, it introduces complexity that overwhelms the model, leading to interference and degraded generalization. At the benchmark level, these moderate gains are distributed across AIME-25, Math 500, and LMGH, with only minor changes on GPQA and GMQ.

In contrast, the official Qwen3-1.7B model benefits more steadily from reverse augmentation, improving from 27.4% to 28.7% with two subsets and reaching 29.2% with four. Despite minor fluctuations, the overall trend suggests that higher-capacity, post-trained models are more capable of leveraging larger amounts of reverse supervision, where increased directional diversity provides complementary signals that enhance generalization. Notably, the strongest improvements occur on AIME-25 and Math500, while GPQA, GMQ, and LMGH remain relatively stable. These results suggest that the effective utilization of reverse data at larger scales is contingent on sufficiently capable and well-trained models.

## 5 CONCLUSION

This work explores reasoning alignment through reverse data augmentation and introduces Latent-DPO, an extension of Direct Preference Optimization designed to preserve shared reasoning knowledge while enforcing direction-aware separation. Leveraging curated exemplars, we construct reverse data through a straightforward pipeline, achieving quality that matches the original LIMO dataset. Experiments show that 817 reverse examples improve base performance by 4.5% on average across five benchmarks. Ablations confirm the critical roles of the latent variable and Uniform-KL regularization, with the removal of either reducing accuracy by 1.1%–4.7%. Scaling studies show that Latent-DPO achieves gains of 0.9%–3.2%, confirming that reverse data consistently enhances performance: larger datasets bring greater benefits to higher-capacity models, whereas excessive augmentation can degrade performance in less capable models. In summary, reverse data augmentation and Latent-DPO jointly prove effective for reasoning alignment, offering a data-efficient means of improving both alignment quality and performance across diverse domains.

## 6 REPRODUCIBILITY STATEMENT

We have taken extensive measures to ensure the reproducibility of our work. The reverse-augmented dataset along with all Latent-DPO code and training scripts are released in an anonymous repository: `https://anonymous.4open.science/r/submission_429`. Detailed training configurations, including hyperparameters, optimizer settings, and hardware resources, are provided in Section 4.1 and Appendix D. Our evaluation strictly follows the `lm-eval-harness` framework (Gao et al., 2024), ensuring transparent and consistent comparisons across benchmarks. Additional ablations, implementation details, and dataset construction procedures are included in Appendix C to further support full reproducibility.

## 7 ETHICS STATEMENT

This work aims to advance reasoning alignment in large language models through data-efficient augmentation and Latent-DPO. By promoting greater consistency in reasoning, our approach may help foster the development of safer and more reliable artificial intelligence systems. However, as with any alignment method, risks of misuse remain, particularly if such systems are applied in sensitive decision-making contexts without appropriate safeguards. All experiments are based on the LIMO mathematical reasoning benchmark with reverse data generated through our pipeline, where the mathematical focus ensures negligible risk of harmful content. We emphasize that our methods are intended solely for responsible use in responsible research.

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

## A  ADDITIONAL RELATED WORK

**High-quality reasoning data and efficient scaling.**    A consistent observation in reasoning align-ment is that the quality and structural diversity of supervision often outweigh raw scale. Less is More for Reasoning (LIMO) shows that only 817 curated exemplars can significantly improve reasoning performance and generalization (Ye et al., 2025). Similar results are observed in efficiency-oriented scaling approaches, where compact models, when paired with high-quality exemplars, can perform comparably to much larger systems (Muennighoff et al., 2025). Beyond dataset design and model scaling, inference-time diversification methods such as Chain-of-Thought (Wei et al., 2022), self-consistency (Wang et al., 2022), Tree-of-Thoughts (Yao et al., 2023a), Graph-of-Thoughts (Besta et al., 2024), and Program-of-Thoughts (Chen et al., 2022), along with tool-augmented strategies like ReAct (Yao et al., 2023b), Toolformer (Schick et al., 2023), and PAL (Gao et al., 2023), further confirm that aggregating diverse trajectories enhances robustness. On the training side, STaR self-distillation (Zelikman et al., 2022), Self-Refine (Madaan et al., 2023), verifier filtering (Cobbe et al., 2021), and feedback methods such as Constitutional AI (Bai et al., 2022b; Ouyang et al., 2022) demonstrate that well-structured supervision can substitute for large-scale brute force. However, most approaches remain limited to *forward-only* supervision, rely on powerful teacher models, or demand costly filtering for high-quality data.

## B  LATENT VARIABLE PREFERENCE OPTIMIZATION WITH DIRECTION CONSISTENCY

### B.1  FROM STANDARD DPO TO DIRECTION-AWARE FORMULATION

Given a prompt $x$ and a preferred–dispreferred pair $(y^+, y^-)$, standard DPO Rafailov et al. (2023) optimizes

$$p_\theta(y^+ \succ y^- \mid x) = \sigma\big(\beta\, m_\theta(x, y^+, y^-)\big), \tag{7}$$

where $\beta > 0$ and $m_\theta$ is the log-likelihood margin between $y^+$ and $y^-$ under the policy and reference model.

In structured reasoning tasks, $y^-$ may still follow the same reasoning trajectory as $y^+$ (*direction-consistent*) but contain minor slips. Penalizing such cases as harshly as direction-divergent ones can hinder generalization. We therefore introduce binary latent variables $z_w, z_l \in \{0, 1\}$ indicating whether $y^+$ and $y^-$ are direction-consistent with $x$.

### B.2  POSTERIOR PARAMETERIZATION

We define a *soft* posterior

$$q_\phi(z \mid x, y) = \text{softmax}\big(u_\phi(x, y)\big), \quad u_\phi : \mathbb{R}^d \to \mathbb{R}^2,$$

using the hidden state of the *last non-padding token* from $(x; y)$ as input. The posterior probabilities

$$q_w \triangleq q_\phi(z{=}1 \mid x, y^+), \quad q_l \triangleq q_\phi(z{=}1 \mid x, y^-)$$

share the same classifier for calibration and parameter efficiency.

### B.3  POSTERIOR-WEIGHTED MARGIN

Let $\ell_\theta(\cdot \mid x)$ and $\ell_\text{ref}(\cdot \mid x)$ denote the sum of token log-likelihoods (answer tokens only) under the trainable policy and frozen reference model, respectively. We construct the posterior-weighted margin:

$$m_\text{eff} = \big[\, q_w\, \ell_\theta(y^+ \mid x) - (1 - q_l)\, \ell_\theta(y^- \mid x) \,\big] - \big[\, q_w\, \ell_\text{ref}(y^+ \mid x) - (1 - q_l)\, \ell_\text{ref}(y^- \mid x) \,\big]. \tag{8}$$

Here $q_w$ amplifies preferred responses deemed direction-consistent, while $(1 - q_l)$ suppresses neg-atives predicted as direction-divergent. When $(q_w, q_l) \to (1, 0)$, Eq.8 recovers the standard DPO margin.

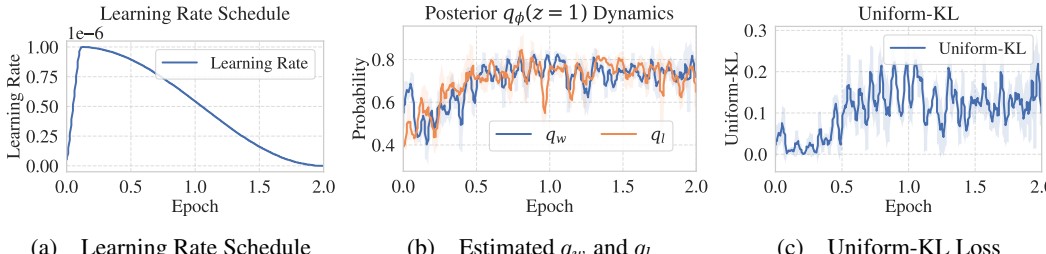

(a) Learning Rate Schedule    (b) Estimated $q_w$ and $q_l$    (c) Uniform-KL Loss

Figure 4: Training dynamics of Latent-DPO. Panel (a) illustrates the learning rate schedule over training steps. Panel (b) shows the estimated alignment posteriors $q_w$ and $q_l$, which act as soft gates controlling the effective DPO margin. Panel (c) tracks the Uniform-KL regularization loss, ensuring that the posterior does not collapse prematurely. Shaded regions indicate variability across runs.

### B.4 VARIATIONAL INFERENCE PERSPECTIVE

This formulation can be derived from a latent-variable ELBO under the Bradley–Terry preference model:

$$\mathcal{L}_{\mathrm{ELBO}} = \mathbb{E}_{q_\phi(z_w, z_l)}[-\log \sigma(\beta m(z_w, z_l))] + \lambda_{\mathrm{KL}} \, \mathrm{KL}\big(q_\phi(z_w, z_l) \, \| \, p(z_w, z_l)\big), \qquad (9)$$

where $p(z_w, z_l)$ is a prior over direction-consistency. In our main configuration, we adopt a *uniform prior* $p(z_w, z_l) = \mathrm{Uniform}(\{0,1\}^2)$ with $\lambda_{\mathrm{KL}} > 0$, which encourages entropy in the latent posterior and prevents premature collapse. For ablation, we remove the KL regularization by setting $\lambda_{\mathrm{KL}} = 0$, reducing Eq. 9 to the Jensen surrogate:

$$\mathcal{L}_{\mathrm{sur}} = -\log \sigma\big(\beta \, m_{\mathrm{eff}}\big). \qquad (10)$$

The Jensen gap is bounded by $\frac{\beta^2}{8} \mathrm{Var}_{q_\phi}[m]$ and vanishes as $q_\phi$ concentrates, ensuring that Eq. 10 remains a principled variational approximation. Ablations without KL regularization ($\lambda_{\mathrm{KL}} = 0$) are reported in Table 3.

### B.5 IMPLEMENTATION IN EXPERIMENTS

- **Hidden representation:** last non-padding token state from $(x; y)$.
- **Posterior head:** shared two-layer MLP with softmax over two classes.
- **Weighting:** symmetric application of $(q_w, 1 - q_l)$ to both policy and reference terms.
- **Main setting:** uniform prior with KL-to-uniform regularization ($\lambda_{\mathrm{KL}} > 0$).
- **Ablation:** remove KL regularization ($\lambda_{\mathrm{KL}} = 0$).

### B.6 TRAINING DYNAMICS

Figure 4 presents the training dynamics of Latent-DPO. Although the smoothed epoch-level trajectories of $q_w$ and $q_l$ appear close, Latent-DPO uses $q_w$ and $1 - q_l$ as a probabilistic gate on the preference margin, allowing the effective margin $m_{\mathrm{eff}}$ to be modulated softly rather than enforced through hard separation.

## C PROMPT TEMPLATES AND GENERATION SETTINGS

We use the following decoding and sampling parameters throughout data generation:

```
"messages": messages,
"think_budget": 8192,
"max_tokens": 2048,
"temperature": 0.6,
"top_p": 0.95,
"top_k": 30,
"stream": False,
```

**Prompt templates.**  For reverse problem generation, we adopt the following template:

```
Given:
Original Problem:
{question}

Original Answer: {answer}

Your task:
Create 3 reverse problems inspired by this original problem.
Each reverse problem must:
1. Be fully specified and no hidden or missing conditions
2. Have exactly one unique correct answer, with a clear reason for its
   uniqueness
3. Connect meaningfully to the original problem by changing known vs
    unknown,
   modifying parameters, or extending constraints
4. Use a distinct reasoning path (e.g. modular arithmetic, factorization,
   parity arguments, number-theoretic decomposition, series summation)
5. Be rigorous, consistent, and suitable for advanced learners
6. Explicitly check that no smaller or alternative solution exists
   to verify uniqueness

Return four problems in this format:

Problem 1
- Statement:
- Answer:
- Sketch:

Problem 2
- Statement:
- Answer:
- Sketch:

Problem 3
- Statement:
- Answer:
- Sketch:
```

For solution generation, we apply the following template:

```
System: You are a helpful reasoning assistant.
User: {question}
```

Finally, for verification we use a fact-checking prompt:

```
You are a meticulous fact-checking assistant.
1. Carefully reason through the answer of the question.
2. You may cite relevant facts, knowledge, or perform calculations to
    support your analysis.
3. Once you reach a conclusion, output exactly two clean marker lines as
   follows:
   - JUDGE: <yes|no>
      'yes' if the model's verdict is factually correct, 'no' otherwise.
Question: {question}
Model verdict (yes/no): {model_ans}
```

**Examples of Reverse Problem Construction**   To illustrate how reverse problems are constructed from original forward problems, we provide below one representative case.

**Original problem:** Circles of radius 3 and 6 are externally tangent to each other and are internally tangent to a circle of radius 9. The circle of radius 9 has a chord that is a common external tangent of the other two circles. Find the square of the length of this chord.

**Original answer:** 224.

**Reverse problems:**

1. *A circle of radius 9 contains two smaller circles that are internally tangent to it and externally tangent to each other. A chord of the large circle is a common external tangent to the two smaller circles and has length $4\sqrt{14}$. Determine the radii of the two smaller circles.*

2. *A circle of radius 9 contains two smaller circles of radii $r$ and $2r$ that are internally tangent to it and externally tangent to each other. A chord of the largest circle is a common external tangent to the smaller circles. If the square of the chord length is 224, find $r$.*

3. *Two circles of radii 3 and 6 are externally tangent to each other and internally tangent to a larger circle. A chord of the larger circle is a common external tangent to the smaller circles. If the square of the chord length is 224, find the radius of the larger circle.*

These examples demonstrate how reverse supervision is systematically constructed: each reverse problem maintains a close semantic link to the original forward problem while introducing a new perspective (e.g., altering the unknown, parameterizing radii, or changing tangency relations).

Table 5: Reverse Data Quality Evaluation based on Qwen3-1.7B-Base.

| Dataset | Accuracy | AIME-25 | GPQA | Math 500 | GMQ | LMGH | Avg. Acc. |
|---------|----------|---------|------|----------|-----|------|-----------|
| Subset 1 | - | 3.3% | 30.3% | 46.0% | 33.6% | 12.1% | 25.1% |
| Subset 2 | 57.4% | 6.7% | 30.8% | 46.6% | 33.6% | 13.6% | 26.3% |
| Subset 3 | 60.7% | 3.3% | 28.3% | 47.0% | 37.3% | 11.4% | 25.5% |

Table 6: Reverse Data Quality Evaluation based on Qwen3-1.7B.

| Dataset | Accuracy | AIME-25 | GPQA | Math 500 | GMQ | LMGH | Avg. Acc. |
|---------|----------|---------|------|----------|-----|------|-----------|
| Subset 1 | - | 6.7% | 26.8% | 32.2% | 32.5% | 9.9% | 21.6% |
| Subset 2 | 57.4% | 10.0% | 30.3% | 30.4% | 34.2% | 7.6% | 22.5% |
| Subset 3 | 60.7% | 3.3% | 29.3% | 33.6% | 31.3% | 5.3% | 20.6% |

Table 7: Effect of Mixing Forward (LIMO) and Reverse Subsets on Distillation Performance.

| Dataset | AIME-25 | GPQA | Math 500 | GMQ | LMGH | Avg. Acc. |
|---------|---------|------|----------|-----|------|-----------|
| LIMO + Subset 1 | 0% | 30.3% | 47.4% | 33.1% | 7.6% | 23.7% |
| LIMO + Subset 1 and 2 | 0% | 30.3% | 40.2% | 35.6% | 10.6% | 23.3% |
| LIMO + Subset 1, 2 and 3 | 0% | 29.3% | 43.4% | 33.6% | 7.6% | 22.8% |

### C.1 MULTI-RUN ROBUSTNESS OF LATENT DPO

To assess stability, we repeat Latent DPO training three times for each base model family (Qwen3-1.7B-Base, SFT(Mixed), and Qwen3-1.7B), reporting mean accuracy and standard deviation across runs. Results are shown in Table 8. The small variance across runs suggests that Latent DPO yields consistent gains independent of random initialization and training noise.

## D IMPLICATION DETAILS

We implement all experiments within the SWIFT framework (Zhao et al., 2024). Supervised fine-tuning (SFT) is performed on the Qwen3-1.7B-Base model using the standard SWIFT framework, trained for 3 epochs with a learning rate of $1 \times 10^{-5}$. Latent-DPO is implemented by extending the framework with a custom LatentDPOLoss module that augments the DPO objective with posterior modulation and Uniform-KL regularization. Training is performed for 3 epochs with a

Table 8: Statistical experiment information of Latent DPO averaged over three independent runs.

| Base Model | AIME-25 | GPQA | Math 500 | GMQ | LMGH |
|---|---|---|---|---|---|
| Qwen3-1.7B-Base | $4.4\% \pm 1.9\%$ | $27.8\% \pm 0\%$ | $46.6\% \pm 0.2\%$ | $36.7\% \pm 0.7\%$ | $2.3\% \pm 0.8\%$ |
| SFT (Mixed) | $6.7\% \pm 3.3\%$ | $28.5\% \pm 1.2\%$ | $47.4\% \pm 0.6\%$ | $35.8\% \pm 0.4\%$ | $14.0\% \pm 2.7\%$ |
| Qwen3-1.7B (Official) | $15.5\% \pm 1.2\%$ | $28.5\% \pm 0.8\%$ | $47.5\% \pm 1.5\%$ | $36.7\% \pm 0.3\%$ | $13.9\% \pm 2.7\%$ |

learning rate of $1 \times 10^{-6}$, per-device batch size 1, and gradient accumulation 1 (global batch size of 4). All experiments employ DeepSpeed ZeRO-3 (Rajbhandari et al., 2020) in bfloat16 precision for memory efficiency, running on 4×RTX 4090 (24GB) GPUs. For SFT, the maximum sequence length is set to 11,000 tokens to capture the full reasoning traces distilled from the teacher model. For Latent-DPO training, we compare three initialization settings: (1) continuing from the SFT-distilled model, (2) training directly on Qwen3-1.7B-Base, and (3) training directly on Qwen3-1.7B. Depending on the dataset, training takes between one and four hours. In all Latent-DPO cases, the maximum sequence length is capped at 1,000 tokens to balance long-form coverage with the computational feasibility of preference optimization.

## E  USE OF LARGE LANGUAGE MODELS

During manuscript preparation, a large language model (LLM) was used occasionally as an auxiliary tool to refine language, including improving fluency and readability. The LLM was not involved in generating original research contributions: it did not formulate research questions, design methodologies, conduct experiments, analyze results, or draft substantive scientific content. All core intellectual work, such as idea development, experimental execution, and interpretation of findings, was carried out independently by the authors. Any linguistic suggestions from the LLM were critically reviewed and selectively adopted to ensure accuracy, originality, and scholarly integrity. The authors take full responsibility for the research content and conclusions, and the LLM is not listed as a contributor or author.

## F  REVERSE DATA QUALITY DETAILS

**Reverse Data Distillation.**  Table 5 reports the performance of SFT models distilled on each reverse subset individually. Across all benchmarks, every reverse subset outperforms distillation on the forward-only LIMO data, confirming that inversion supervision provides stronger alignment signals. Subset 1, which was further filtered for correctness, does not yield clear improvements over the unfiltered subsets. Notably, Subset 2 achieves the highest average accuracy (26.26%), demonstrating that reverse data can be reliably beneficial even without strict filtering. In contrast, Table 6 shows that applying additional distillation on models that have already undergone Reinforcement Learning leads to a marked degradation in performance.

**Mixing Forward and Reverse Data.**  Table 7 compares models distilled on mixtures of the forward LIMO dataset and its reverse counterparts. Here we observe that mixing forward and reverse data does not lead to performance gains over single-direction SFT. Instead, unstructured combinations tend to introduce interference, reducing the average accuracy compared to training on either forward or reverse data alone. This suggests that naive mixing is not synergistic and motivates the need for direction-aware preference optimization strategies.

## G  TWO CASE STUDIES DEMONSTRATING LATENT-DPO'S IMPROVEMENT IN REASONING ABILITY AND CHAIN QUALITY

In this section, we present two case studies to demonstrate how Latent-DPO improves reasoning ability and chain quality. We will compare the outputs of different models and discuss their reasoning process in detail.

**Case 1:** Find the sum of all positive integers $n$ such that $n+2$ divides the product $3(n+3)(n^2+9)$.

*Latent-DPO trained based on Qwen3-1.7B-Base's Answer Chain:*

1. **Divisibility Condition:** We need $n + 2$ to divide $3(n + 3)(n^2 + 9)$. This implies that
$$3(n + 3)(n^2 + 9) \equiv 0 \pmod{n + 2}.$$

2. **Simplify the Expression and Evaluate Polynomial:** We use the Remainder Theorem to evaluate $3(n + 3)(n^2 + 9)$ at $n = -2$, which yields:
$$3(-2 + 3)((-2)^2 + 9) = 3(1)(4 + 9) = 3(1)(13) = 39.$$

3. **Divisors of 39:** The divisors of 39 are $1, 3, 13, 39$. Therefore, $n + 2$ must be one of these, giving possible values for $n$ as $1, 11$, and $37$.

4. **Sum Valid Values:** The sum of the valid $n$-values is:
$$1 + 11 + 37 = 49.$$

*DPO Answer Chain:*

1. **Rotation Transformation:** Correct transformation applied.
2. **Substitute and Simplify:** Struggles to simplify and solve the intersection system.
3. **Solve for Intersection:** Incorrect extraction of the intersection point.
4. **Answer:** Incorrect final answer, outputting 10.

*Qwen3-1.7B-Base Answer Chain:*

1. **Rotation Transformation:** Correct transformation applied.
2. **Substitute and Simplify:** truggles to simplify the resulting expressions.
3. **Solve for Intersection:** Fails to correctly solve the system of equations.
4. **Answer:** Incorrect final answer, outputting 10.

The base and DPO models struggle with simplifying the expressions and fail to reach the correct solution, outputting an incorrect answer of 10. In contrast, latent-DPO excels in reasoning ability and chain quality by accurately applying the divisibility condition and simplifying complex expressions using the Remainder Theorem. It efficiently handles polynomial division and identifies the correct divisors, ultimately summing the valid values to produce the correct answer of 49.

**Case 2:** Find the sum of all integer bases $b > 9$ for which $17_b$ is a divisor of $97_b$.

*Latent-DPO trained based on Qwen3-1.7B-Base's Answer Chain:*

1. **Convert Base $b$ Numbers to Base 10:**
   - $17_b = b + 7$
   - $97_b = 9b + 7$

2. **Divisibility Condition:** We need $b + 7$ to divide $9b + 7$. This means:
$$9b + 7 \equiv 0 \pmod{b + 7}$$
   Simplifying the expression:
$$9b + 7 = 9(b + 7) - 56 \equiv -56 \equiv 0 \pmod{b + 7}$$
   Thus, $b + 7$ must be a divisor of 56.

3. **Divisors of 56:** The divisors of 56 are $1, 2, 4, 7, 8, 14, 28, 56$. Since $b > 9$, we only consider the divisors greater than 9, which are $14, 28, 56$.

4. **Valid Bases:** From the divisors $14, 28, 56$, we subtract 7 to get the corresponding values for $b$:
$$b = 14 - 7 = 7, \quad b = 28 - 7 = 21, \quad b = 56 - 7 = 49$$
   Since we require $b > 9$, the valid bases are 21 and 49.

5. **Sum of Valid Bases:** The sum of the valid bases is:

$$21 + 49 = 70$$

*DPO Answer Chain:*

1. **Divisibility Condition Correctly Identified:** The model correctly identified the requirement that $b + 7$ must divide $9b + 7$, but failed in applying this condition correctly.

2. **Divisors of 56 Correctly Listed:** The model correctly listed the divisors of 56 but incorrectly included bases corresponding to these divisors without filtering for the condition $b > 9$.

3. **Incorrect Final Set of Valid Bases:** The model incorrectly included bases such as 16, 25, 36, and others, which are not valid since they do not satisfy the condition $b + 7$ divides $9b + 7$.

*Qwen3-1.7B-Base Answer Chain:*

1. **Divisibility Condition:** Correctly identifies that $b + 7$ must divide $9b + 7$.

2. **Simplify the Expression:** Correctly derives the equation $b = \frac{7(k-1)}{9-k}$, and simplifies the equation correctly.

3. **Solve for $b$:** Correctly finds possible values of $k$, but *incorrectly includes invalid bases* that do not satisfy $b > 9$. This led to an invalid sum of 275.

4. **Answer:** Incorrect final answer of 275, instead of the correct sum of 70. The model fails to properly filter the valid bases, missing bases $b = 21$ and $b = 49$.

## H  PERFORMANCE COMPARISON OF LATENT-DPO WITH OTHER DPO VARIANTS

As shown in Table 9, Latent-DPO demonstrates significant improvements over $\beta$-DPO, $\gamma$-DPO and Simpo in the unaligned base model. For the RL-aligned Qwen3-1.7B, both Latent-DPO and $\beta$-DPO exhibit distinct advantages. $\beta$-DPO performs well on tasks like GPQA and GMQ, while Latent-DPO outperforms in handling more complex reasoning tasks, such as Math 500 and LMGH, where its ability to manage intricate reasoning steps is particularly effective.

Table 9: Performance comparison of Latent-DPO with other DPO variants ($\beta$-DPO Wu et al. (2024), $\gamma$-DPO Sun et al. (2025), and SimPO Meng et al. (2024)) across different models.

| Base Model | DPO-Variants | AIME-25 | GPQA | Math 500 | GMQ (Agieval) | LMGH | Avg. Acc. |
|---|---|---|---|---|---|---|---|
| Qwen3-1.7B-Base | $\beta$-DPO | 0% | 27.3% | 47.6% | 35.6% | 3.8% | 22.9% |
| Qwen3-1.7B-Base | $\gamma$-DPO | 0% | 28.3% | 45.6% | 35.0% | 1.5% | 22.1% |
| Qwen3-1.7B-Base | SimPO | 0% | 27.8% | 47.0% | 33.3% | 3.0% | 22.2% |
| Qwen3-1.7B-Base | Ours | 6.7% | 27.8% | 46.8% | 34.8% | 3.8% | 24.0% |
| Qwen3-1.7B | $\beta$-DPO | 16.7% | 32.8% | 47.0% | 36.8% | 10.6% | 28.7% |
| Qwen3-1.7B | $\gamma$-DPO | 20.0% | 31.3% | 46.4% | 35.3% | 7.6% | 28.1% |
| Qwen3-1.7B | SimPO | 13.3% | 28.8% | 45.6% | 35.9% | 13.6% | 27.4% |
| Qwen3-1.7B | Ours | 16.7% | 28.3% | 49.2% | 35.9% | 13.6% | 28.7% |

## I  HUMAN VERIFICATION OF REVERSE DATA QUALITY

We evaluated the dataset through a manual assessment, focusing on the completeness, clarity, organization, and logical coherence of the reasoning, as well as the level of detail and task difficulty. Two PhD-level researchers, each with expertise in data science, independently reviewed 50 randomly selected verified reverse reasoning samples. Each sample was assessed to ensure the reasoning steps were clear, logically organized, and complete. The analysis confirmed the dataset's reliability for training models on complex tasks. In terms of difficulty, the tasks were categorized as moderate and high. 21% of the samples were classified as moderate, with straightforward reasoning, while

79% were high difficulty, requiring advanced, multi-step reasoning. This distribution underscores the dataset's emphasis on challenging tasks, making it well-suited for evaluating models designed for sophisticated reasoning.

