# OpenReview forum: "Latent-DPO: Direction-Aware Preference Optimization for Reasoning Alignment"
_ICLR.cc/2026/Conference — Submitted to ICLR 2026_

### Official Review · Reviewer_3ALD · 2025-10-18

**Soundness:** 2
**Presentation:** 2
**Contribution:** 2
**Rating:** 4
**Confidence:** 3

**Summary:**

The paper proposes Latent-DPO, a variant of Direct Preference Optimization that introduces a binary latent variable to indicate whether a response follows the intended reasoning direction. The method is evaluated using forward-and-reverse pairs automatically built from the 817-problem LIMO dataset.

**Strengths:**

1. The writing is clear and easy to follow
2. They provided the code, improving the reproducibility of the work

**Weaknesses:**

1. The paper seems to lack novelty. Reverse reasoning and DPO are not new concepts and the combination does not seem to yield a particularly notable gain.
2. The experiments are conducted on Qwen3-1.7B-Base and Qwen3-1.7B alone. The paper might benefit from more experiments on 7B, 8B, and 32B models.
3. The improvements seem slight compared to other methods and even the base model, especially considering that some of the benchmarks have very limited samples.

**Questions:**

See weaknesses.

---

> ### Author Response · Authors · 2025-11-22
> **Novelty and Additional Experiments**
>
> Thank you for your insightful and constructive review. We have carefully addressed each point and made the necessary revisions in the revised manuscript, which accompanies this response. Below are our responses to your concerns.
>
> **Q1: Novelty**
>
> While reverse reasoning and DPO are established concepts, our approach introduces a novel mechanism for enhancing reasoning alignment by modeling direction consistency through a latent variable. Instead of naively applying DPO to reverse reasoning, we use the model's inherent belief to dynamically adjust the DPO margin based on reasoning-path consistency. This allows us to preserve shared reasoning knowledge while separately accounting for divergent reasoning directions, a challenge that has not been adequately addressed in prior work. By incorporating this dynamic adjustment mechanism, Latent-DPO ensures more coherent and consistent reasoning across both forward and reverse directions, leading to improved overall performance.
>
> For example, as illustrated in Figure 1 of the paper, in a rectangle-area problem, DPO might treat the forward operation (multiplication) and the reverse operation (division) as entirely separate, effectively disregarding the shared reasoning structure and considering it a negative example. In contrast, Latent-DPO applies mild adjustments to subtle differences, such as small shifts in reasoning direction, while maintaining the shared knowledge between the positive and negative directions. In our example, Latent-DPO seeks to preserve the shared formula for area (used in both forward and reverse directions) while still differentiating between multiplication and division, leading to more coherent and consistent reasoning across both directions.
>
> **Q2: Additional experiments**
>
> **Table 1   Supplementary Experiments with Qwen3-4B as the Baseline**
> |**Base Model**|**Report**|**AIME-25**|**GPQA**|**Math 500**|**GMQ**|**LMGH**|**Avg. Acc.**|
> |-|-|-|-|-|-|-|-|
> |Qwen3-4B|Released-Model|10%|40.4%|49.2%|47%|35.6%| 36.4%|
> |Qwen3-4B|Latent-DPO(Exp1)|26.7%|41.4%| 50.4%|46.7%| 31.2%| 39.3%|
> |Qwen3-4B|Latent-DPO(Exp2)|20%|38.4%| 51.6%| 44.7%| 40.9%| 39.1%|
> |Qwen3-4B|Latent-DPO(Exp3)|30%|38.4%|49%| 45.6%| 34.1%|39.4%|
> |Evaluation|Exp1-3|25.6% ± 5.5%|39.4% ± 1.6%|50.3% ± 1.1%|45.7% ± 1%|35.4% ± 4.9%|39.3% ± 0.1%|
>
> Note: In this table, the "±" represents the standard deviation of the accuracy, indicating the variability in the performance of the model across three independent runs. Exp1-3 refers to three independent experiments conducted under identical settings. Each experiment was run three times with varying initial seeds to assess the model's robustness and consistency.
>
> We have expanded our experiments by including the Qwen3-4B model as an additional baseline, as shown in Table 1. Latent-DPO demonstrates clear improvements to both in-domain and near-domain tasks, particularly on AIME-25 and Math 500. For AIME-25, Latent-DPO improves significantly from **10%** to **26.7%**, while for Math 500, it increases from **49.2%** to **50.4%**. Latent-DPO shows some variability in performance on out-of-domain benchmarks, but the overall average demonstrates an improvement over the base model. Despite the limited size of the reverse-reasoning dataset, Latent-DPO consistently delivers improvements across most benchmarks across three independent runs, highlighting its robustness. Our experiments with Qwen3-4B, Qwen3-1.7B-Base, Qwen3-1.7B, and SFT model further confirm Latent-DPO's capability, with consistent gains observed across different model variants.
>
> **Table 2: Performance Comparison of Latent-DPO and Other DPO Variants**
>
> | **Base Model**|**DPO-Variants**|**AIME-25**|**GPQA**|**Math 500**|**GMQ (Agieval)**|**LMGH**|**Avg. Acc.**|
> |-|-|-|-|-|-|-|-|
> | Qwen3-1.7B-Base| β-DPO| 0%| 27.3%| 47.6%| 35.6%  | 3.8% | 22.9%|
> | Qwen3-1.7B-Base| γ-DPO| 0%  | 28.3%| 45.6% | 35.0% | 1.5%| 22.1%|
> | Qwen3-1.7B-Base| SimPO| 0% | 27.8% | 47%| 33.3%| 3.0%| 22.2%|
> | Qwen3-1.7B-Base| Ours  | 6.7%| 27.8%| 46.8%| 34.8%|3.8%| 24.0%|
> | Qwen3-1.7B | β-DPO |16.7%| 32.8%| 47%| 36.8%|10.6%|28.7%|
> | Qwen3-1.7B| γ-DPO| 20.0%| 31.3% | 46.4%| 35.3%|7.6% | 28.1%|
> | Qwen3-1.7B| SimPO| 13.3%| 28.8%| 45.6%| 35.9%|13.6%|27.4%|
> | Qwen3-1.7B| Ours| 16.7%| 28.3%| 49.2%| 35.9%|13.6%|28.7%|
>
> As shown in Table 2, Latent-DPO demonstrates significant improvements over β-DPO, γ-DPO and simpo in the unaligned base model. For the RL-aligned Qwen3-1.7B, both Latent-DPO and β-DPO exhibit distinct advantages. β-DPO performs well on tasks like GPQA and GMQ, while Latent-DPO outperforms in handling more complex reasoning tasks, such as Math 500 and LMGH, where its ability to manage intricate reasoning steps is particularly effective.

---

> ### Author Response · Authors · 2025-11-22
> **Gains with Limited Data**
>
> **W3: Gains with Limited Data**
>
> Limited samples: we acknowledge that AIME-25 is relatively small, but its complexity and challenging nature make it a highly relevant benchmark for assessing reasoning capabilities. Math-500, with its diverse set of mathematical problems across various difficulty levels, provides a rigorous evaluation of our method’s effectiveness. Additionally, benchmarks like GPQA, GMQ, and LMGH are well-established in the field and widely recognized for their value in testing generalization and reasoning across different domains, ensuring the reliability of our evaluation standards.
>
> Slight gains: as shown in Table 2 and Table 4 of the paper, the experiments based on SFT (Mixed) based model and training with two sets of reverse-reasoning data on the base model demonstrate consistent and notable improvements across all benchmarks. While the performance gains for Qwen3-1.7B and Qwen3-4B in Table 1 are most pronounced on AIME-25 and Math 500, it is important to note that these models were RL-aligned and then enhanced by Latent-DPO. Despite using only 817 reverse-reasoning samples, Latent-DPO achieves substantial improvements, particularly in in-domain tasks.
>
> We sincerely thank the reviewers for their valuable feedback. We believe the revisions have addressed all concerns and strengthened the manuscript. We hope these changes improve the quality of the paper and look forward to your feedback.

---

> > ### Comment · Reviewer_3ALD · 2025-11-26
> >
> > Thank you for the clarifications. However, I am still a little concerned about the scalability and the upper bound of the method. Whether it can be generalized to a stronger reasoning model is also unclear, which is crucial in demonstrating its potential to achieve higher accuracy. Therefore, I would keep my original rating.

---

### Official Review · Reviewer_YKRc · 2025-10-24

**Soundness:** 2
**Presentation:** 2
**Contribution:** 2
**Rating:** 4
**Confidence:** 5

**Summary:**

This paper introduces Latent-DPO, an extension of Direct Preference Optimization that incorporates a binary latent variable to model reasoning-path consistency, aiming to preserve shared knowledge while adaptively enforcing separation between forward and reverse reasoning directions in language models. The key contributions are: (1) Reverse Data Augmentation: A method to construct high-quality reverse reasoning questions from a small seed dataset (LIMO), demonstrating that even 817 reverse examples can improve model performance. (2) Latent Direction Alignment: A new algorithm that introduces a binary latent variable to model whether a response follows the intended reasoning path. This allows the model to adaptively modulate the DPO loss, aiming to preserve shared reasoning knowledge while enforcing separation only on direction-specific operations.

**Strengths:**

(1) Problem Motivation: The paper identifies a potentially interesting and nuanced problem: that standard DPO might be too blunt an instrument for reasoning tasks where "dispreferred" responses may still contain valuable, shared knowledge.

(2) Theoretical Formulation: The latent-variable framework is derived with mathematical care, and the connection to variational inference is a positive aspect of the work.

(3) Empirical Effort: The authors have undertaken a non-trivial empirical study, constructing a reverse dataset and running multiple experiments across different model initializations.

**Weaknesses:**

(1) Misalignment Between Motivation and Method: The core motivation is to preserve shared knowledge from "partially correct" responses. However, the construction of the preference pairs (x, y+, y-) is problematic. The y- for a forward problem can be a correct solution to the reverse problem, not a partially correct one. The latent variable q_l is trained to detect a different reasoning direction, not a flawed one. This fundamentally undermines the proposed mechanism's ability to achieve its stated goal of preserving shared knowledge from incorrect but structurally similar responses. The y- examples are a mix of directionally incorrect but otherwise valid solutions and potentially fully wrong answers, creating a noisy and inconsistent learning signal for the latent classifier.

(2) Unjustified Complexity: The method introduces significant complexity: a multi-stage pipeline for reverse data generation and an auxiliary neural network (the posterior head) that requires careful training with a novel regularizer.

(3) Limited Applicability to Non-Invertible Reasoning Tasks: The current approach relies on the construction of well-defined "reverse" problems, however they may not exist for some open-ended reasoning tasks .

**Questions:**

(1) Mechanism Verification: The latent variable z is intended to model "reasoning-path consistency." Can you provide direct evidence that it learns this? For example, please show examples from the validation set where a y- that shares the correct formula but applies it to the wrong direction has a significantly higher q_l than a y- that is complete nonsense.

(2) y- Composition and Ablation: Your y- is a mixture of direction-swapped solutions and potentially incorrect answers. Did you ablate this? What is the performance if y- is only the direction-swapped (but otherwise correct) solution versus only a truly incorrect solution? This would help disentangle what signal the latent variable is actually capturing.

(3) The method is evaluated solely on mathematical reasoning dataset. How would you envision applying this method to more open-ended reasoning tasks (e.g., commonsense QA, code generation) ?

---

> ### Author Response · Authors · 2025-11-21
> **Mechanism Verification and y- Ablation**
>
> Thank you for your insightful and constructive review. We have carefully addressed each point and made the necessary revisions in the revised manuscript, which accompanies this response. Below are our responses to your concerns.
>
> **Q1: Mechanism Verification**
>
> We present 2 case studies based on validation set data, comparing the results of three different models, all using temperature 0 for answer sampling.
>
> Question 1: Find the sum of all positive integers $ n $ such that $ n + 2 $ divides the product $ 3(n + 3)(n^2 + 9) $.
> Latent-DPO trained based on Qwen3-1.7B-Base's Answer Chain:
> 1. Divisibility Condition: We need $n+2$ to divide the product $ 3(n + 3)(n^2 + 9) $.
> 2. Simplify the Expression and Evaluate Polynomial: Using the Remainder Theorem, evaluate the expression at $n=-2$, which yields 39.
> 3. Divisors of 39: The divisors of 39 are 1, 3, 13, and 39. Therefore, $n+2$ must be one of these, giving possible values for n as 1, 11, and 37.
> 4. Sum Valid Values: The sum of the valid n-values is 49.
>
> Qwen3-1.7B-Base Model Answer Chain:
>
> 1. Rotation Transformation: Correct transformation applied.
> 2. Substitute and Simplify: Struggles to simplify the resulting expressions.
> 3. Solve for Intersection: Fails to correctly solve the system of equations.
> 4. Answer: Incorrect final answer, outputting 10.
>
> DPO Answer Chain:
>
> 1. Rotation Transformation: Correct transformation applied.
> 2. Substitute and Simplify: Struggles to simplify and solve the intersection system.
> 3. Solve for Intersection: Incorrect extraction of the intersection point.
> 4. Answer: Incorrect final answer, outputting 10.
>
> The base and DPO models struggle with simplifying the expressions and fail to reach the correct solution, outputting an incorrect answer of 10.  In contrast, latent-DPO excels in reasoning ability and chain quality by accurately applying the divisibility condition and simplifying complex expressions using the Remainder Theorem. It efficiently handles polynomial division and identifies the correct divisors, ultimately summing the valid values to produce the correct answer of 49.
>
> Question 2: Find the sum of all integer bases $b>9$ for which $17_{b}$ is a divisor of $97_{b}$.
>
> Latent-DPO Answer Chain:
>
> 1. Convert Base b Numbers to Base 10
> 2. Divisibility Condition: Identified the correct condition: b+7 must divide 9b+7,  b+7 must divide 56
> 3. Subtracted 7 from the divisors and sum of Valid Bases.
>
> DPO Answer Chain:
>
> 1. Divisibility Condition Correctly Identified
> 2. Divisors of 56 Correctly Listed
> 3. Incorrect Final Set of Valid Bases: mistakenly includes bases like 16, 25, 36, and others, which do not satisfy the divisibility condition. These are invalid bases, and the model fails to properly apply the filtering condition, leading to incorrect bases in the final answer.
>
> Qwen3-1.7B-Base Answer Chain:
>
> 1. Correctly identifies the condition
> 2. Correctly Simplify the Expression
> 3. Finds possible values of k, but includes invalid bases that do not satisfy b>9, resulting in an final answer of 275.
>
> **Q2: y- Ablation**
>
> We modified the training data by replacing the directionally inconsistent y- samples with truly incorrect solutions. Specifically, we took the corresponding directional solution and forcefully altered the values to incorrect ones, then proceeded with training. Additionally, in cases where no numerical solution existed, we partially removed the corresponding response to provide an incorrect answer instead. As shown in Table 1, incorrect responses, while beneficial in specific cases like GPQA, do not yield consistent improvements across all benchmarks. In contrast, directionally inconsistent y- samples lead to better generalization and enhance the model’s ability to address complex reasoning tasks.
>
> **Table 1: y- Ablation**
> | **Base Model**  | **y-**  | **AIME-25** | **GPQA** | **Math 500** | **GMQ (Agieval)** | **LMGH** | **Avg. Acc.** |
> | -| -|- |-|-|-|-|-|
> | Qwen3-1.7B-Base | Incorrect | 0%| 29.3%| 44.6%| 33.1%| 2.3%| 21.9%|
> | Qwen3-1.7B-Base | Directionally Inconsistent | 6.7%| 27.8%| 46.8% | 34.8%| 3.8% | 24.0%|
> | Qwen3-1.7B | Incorrect  | 10%| 30.3%    | 43.4%| 33.3%| 9.9%| 25.4%|
> | Qwen3-1.7B| Directionally Inconsistent | 16.7%| 28.3%| 49.2%| 35.9%| 13.6%| 28.7%|

---

> ### Author Response · Authors · 2025-11-21
> **Clarifications on Reverse Reasoning, Directional Consistency, and Model Complexity in Latent-DPO**
>
> **Q3 & W3:  Limited Applicability to Non-Invertible Reasoning Tasks**
>
> We understand the reviewer’s concern regarding open-ended reasoning tasks where reversibility may not directly apply. In our opinion, reverse reasoning is particularly valuable for addressing challenges posed by the autoregressive nature of Transformer models in complex tasks. In autoregressive generation, the model progresses from the problem to the answer, but it struggles to generalize by reasoning backward from the answer to the problem. By constructing result-oriented questions, reverse reasoning enables the generation of more complex training data, thereby improving the model's robustness and accuracy — particularly in open-ended tasks where forward-only reasoning may not be sufficient. We primarily focus on the example of Code Generation Application to illustrate the effectiveness of reverse reasoning.
>
> 1. Given the output and input data, it allows the model to trace back from the expected output to infer code implementation details, enhancing generation accuracy. For example, when provided with the output and input, the model can deduce the necessary logic for generating the corresponding code segment.
> 2. Step-by-Step Code Logic Validation: Reverse reasoning can also be used to validate the logical correctness of each step in the code generation process. After generating each code snippet, the model checks whether it aligns with the expected logic, helping to prevent errors from accumulating. For instance, when generating a sorting algorithm, the model can backtrack from the final sorted output to verify each sorting step’s correctness, ensuring that every intermediate step contributes correctly to the final result.
> 3. Debugging and Error Localization: In debugging, reverse reasoning helps pinpoint issues within the code. When an incorrect output is produced, the model can trace back to the source, identify the error’s origin, and suggest fixes.
>
> **W1:  Clarification on the Motivation**
>
> Given the high semantic similarity often observed between forward and reverse problems (e.g., as shown in Figure 2 of the paper with Reverse Questions 2 and 3), despite having different reasoning directions, models may struggle to distinguish the consistency of reasoning paths. This discrepancy motivates our approach to model directional consistency and fine-tune the LLM to align these reasoning paths effectively. Although the y- examples are directionally incorrect, they are intentionally selected to help the model differentiate between distinct reasoning paths. By incorporating such directionally divergent examples, we enable the model to better generalize, differentiate reasoning structures, and ultimately improve its ability to handle complex tasks. This approach ensures that the model preserves shared knowledge while effectively distinguishing between valid yet directionally divergent reasoning paths.
>
> **W2:  Unjustified Complexity**
>
> The additional parameters introduced by the posterior head are minimal, contributing only 4–5k parameters for a 1.7B model, which is approximately 0.0003% of the total parameters. This results in a negligible increase in both memory usage and computational overhead, with GPU memory usage rising by just 1–5%, and training time overhead increasing by around 10–20% per step. These additions are not extraneous, but are specifically designed to support the modeling of directional consistency. The multi-stage pipeline for reverse data generation, along with the posterior head, enhances the model’s ability to differentiate reasoning directions and improve generalization, especially in handling complex reasoning tasks.
>
> We sincerely thank the reviewers for their valuable feedback. We believe the revisions have addressed all concerns and strengthened the manuscript. We hope these changes improve the quality of the paper and look forward to your feedback.

---

> > ### Comment · Reviewer_YKRc · 2025-11-28
> >
> > Thank you for the detailed clarification. While I appreciate the potential benefits of reverse reasoning,  I remain somewhat concerned about its broader applicability to truly open-ended reasoning tasks.
> >
> > In many realistic code generation scenarios, the input is an informal natural language description rather than a precise specification, and the expected output may not be uniquely determined or even available at inference time. Under such conditions, applying reverse reasoning (i.e., inferring implementation details from a given output) becomes highly challenging, if not impossible, since the mapping from output back to input logic is often non-invertible or ambiguous.
> >
> > Therefore, while reverse reasoning may enhance performance in controlled or result-supervised settings, its utility appears limited in more general, open-ended contexts where outputs are not known a priori. Given this constraint, I would maintain my original rating.

---

### Official Review · Reviewer_YS8G · 2025-10-25

**Soundness:** 2
**Presentation:** 3
**Contribution:** 2
**Rating:** 4
**Confidence:** 3

**Summary:**

The paper studies direction-aware alignment for reasoning in LLMs, arguing that standard DPO over-separates responses and erodes shared reasoning. It proposes Latent-DPO, which introduces a binary latent variable z to model whether a response is direction-consistent with the prompt and modulates the DPO margin accordingly.

**Strengths:**

- Relative to DPO variants, the key novelty is a latent, learned posterior that adaptively gates margin contributions of win/lose responses based on estimated directional alignment. This is distinct from fixed margins or step-level weighting.
- The reverse data pipeline is not conceptually new, but the careful pairing to form preference tuples and the demonstration that mixed SFT hurts while Latent-DPO helps is a useful empirical insight.
- Overall, contribution is incremental but meaningful: a simple, implementable extension to DPO addressing a concrete failure mode.

**Weaknesses:**

- Limited baseline coverage
- Statistical reporting is thin in main tables. Small gains may be within variance on some benchmarks.
- Potential confound: reverse vs forward stylistic artifacts from different teachers could be learned by the posterior rather than true directional consistency.

**Questions:**

- How does Latent-DPO compare to KTO/SimPO and to step-level/process feedback DPO on the same data?
- Can you report human verification of reverse data quality on a subset, beyond model-based judges?

---

> ### Author Response · Authors · 2025-11-21
> **DPO Variants Performance, Dataset Evaluation, and Supplementary Experiments**
>
> Thank you for your insightful and constructive review. We have carefully addressed each point and made the necessary revisions in the revised manuscript, which accompanies this response. Below are our responses to your concerns.
>
> **Q1: Compare with other DPO Variants**
>
> **Table 1: Performance Comparison of Latent-DPO and Other DPO Variants**
>
> | **Base Model**|**DPO-Variants**|**AIME-25**|**GPQA**|**Math 500**|**GMQ (Agieval)**|**LMGH**|**Avg. Acc.**|
> |-|-|-|-|-|-|-|-|
> | Qwen3-1.7B-Base| β-DPO| 0%| 27.3%| 47.6%| 35.6%  | 3.8% | 22.9%|
> | Qwen3-1.7B-Base| γ-DPO| 0%  | 28.3%| 45.6% | 35.0% | 1.5%| 22.1%|
> | Qwen3-1.7B-Base| SimPO| 0% | 27.8% | 47%| 33.3%| 3.0%| 22.2%|
> | Qwen3-1.7B-Base| Ours  | 6.7%| 27.8%| 46.8%| 34.8%|3.8%| 24.0%|
> | Qwen3-1.7B | β-DPO |16.7%| 32.8%| 47%| 36.8%|10.6%|28.7%|
> | Qwen3-1.7B| γ-DPO| 20.0%| 31.3% | 46.4%| 35.3%|7.6% | 28.1%|
> | Qwen3-1.7B| SimPO| 13.3%| 28.8%| 45.6%| 35.9%|13.6%|27.4%|
> | Qwen3-1.7B| Ours| 16.7%| 28.3%| 49.2%| 35.9%|13.6%|28.7%|
>
> As shown in Table 1, Latent-DPO demonstrates significant improvements over β-DPO, γ-DPO and simpo in the unaligned base model. For the RL-aligned Qwen3-1.7B, both Latent-DPO and β-DPO exhibit distinct advantages. β-DPO performs well on tasks like GPQA and GMQ, while Latent-DPO outperforms in handling more complex reasoning tasks, such as Math 500 and LMGH, where its ability to manage intricate reasoning steps is particularly effective.
>
> **Q2: Human verification of reverse data quality**
>
> We evaluated the dataset through a manual assessment, focusing on the completeness, clarity, organization, and logical coherence of the reasoning, as well as the level of detail and task difficulty. Two PhD-level researchers, each with expertise in data science, independently reviewed 50 randomly selected verified reverse reasoning samples. Each sample was assessed to ensure the reasoning steps were clear, logically organized, and complete.
>
> The analysis confirmed the dataset's reliability for training models on complex tasks. In terms of difficulty, the tasks were categorized as moderate and high. 21% of the samples were classified as moderate, with straightforward reasoning, while 79% were high difficulty, requiring advanced, multi-step reasoning. This distribution underscores the dataset's emphasis on challenging tasks, making it well-suited for evaluating models designed for sophisticated reasoning.
>
> **W1 & W2: Gains with Limited Data**
>
> While the gains are relatively modest on out-of-distribution tasks, Latent-DPO demonstrates notable improvements, especially on in-domain tasks, despite utilizing only 817 reverse-reasoning samples. It is important to highlight that many of the baseline models are already the latest versions and RL-aligned small models. Given this starting point, the observed improvements are even more significant, further emphasizing the effectiveness of Latent-DPO.
>
> **Table 2   Supplementary Experiments with Qwen3-4B as the Baseline**
> |**Base Model**|**Report**|**AIME-25**|**GPQA**|**Math 500**|**GMQ**|**LMGH**|**Avg. Acc.**|
> |-|-|-|-|-|-|-|-|
> |Qwen3-4B|Released-Model|10%|40.4%|49.2%|47%|35.6%| 36.4%|
> |Qwen3-4B|Latent-DPO(Exp1)|26.7%|41.4%| 50.4%|46.7%| 31.2%| 39.3%|
> |Qwen3-4B|Latent-DPO(Exp2)|20%|38.4%| 51.6%| 44.7%| 40.9%| 39.1%|
> |Qwen3-4B|Latent-DPO(Exp3)|30%|38.4%|49%| 45.6%| 34.1%|39.4%|
> |Evaluation|Exp1-3|25.6% ± 5.5%|39.4% ± 1.6%|50.3% ± 1.1%|45.7% ± 1%|35.4% ± 4.9%|39.3% ± 0.1%|
>
> Note: In this table, the "±" represents the standard deviation of the accuracy, indicating the variability in the performance of the model across three independent runs. Exp1-3 refers to three independent experiments conducted under identical settings. Each experiment was run three times with varying initial seeds to assess the model's robustness and consistency.
>
> Additional experiments: we have expanded our experiments by including the Qwen3-4B model as an additional baseline, as shown in Table 2. Latent-DPO demonstrates clear improvements to both in-domain and near-domain tasks, particularly on AIME-25 and Math 500. For AIME-25, Latent-DPO improves significantly from **10%** to **26.7%**, while for Math 500, it increases from **49.2%** to **50.4%**. Latent-DPO shows some variability in performance on out-of-domain benchmarks, but the overall average demonstrates an improvement over the base model. Despite the limited size of the reverse-reasoning dataset, Latent-DPO consistently delivers improvements across most benchmarks across three independent runs, highlighting its robustness. Our experiments with Qwen3-4B, Qwen3-1.7B-Base, Qwen3-1.7B, and SFT model further confirm Latent-DPO's capability, with consistent gains observed across different model variants.

---

> ### Author Response · Authors · 2025-11-21
> **Potential Confound and Human verification of Case Study**
>
> **W3: Potential Confound**
>
> We understand the reviewer’s concern that different teachers may introduce stylistic variations, which could potentially confound the model’s ability to capture true directional reasoning differences. However, it is important to clarify that our approach does not involve using different teacher models for generating the forward and reverse data. Both the forward and reverse reasoning data are generated using the same teacher model, Qwen3-32B. This ensures that the stylistic characteristics remain consistent across both directions of reasoning, effectively addressing the concern that stylistic differences from different teachers could be inadvertently learned by the posterior. By using the same model for both directions, we ensure that the posterior learns true directional consistency based on the reasoning paths themselves, rather than being influenced by stylistic artifacts introduced by different teachers.
>
> **Q2: Human verification of reverse data effectiveness**
>
> We present 2 case studies based on validation set data, comparing the results of three different models, all using temperature 0 for answer sampling.
>
> Question 1: Find the sum of all positive integers $ n $ such that $ n + 2 $ divides the product $ 3(n + 3)(n^2 + 9) $.
>
> Latent-DPO trained based on Qwen3-1.7B-Base's Answer Chain:
> 1. Divisibility Condition: We need $n+2$ to divide the product $ 3(n + 3)(n^2 + 9) $.
> 2. Simplify the Expression and Evaluate Polynomial: Using the Remainder Theorem, evaluate the expression at $n=-2$, which yields 39.
> 3. Divisors of 39: The divisors of 39 are 1, 3, 13, and 39. Therefore, $n+2$ must be one of these, giving possible values for n as 1, 11, and 37.
> 4. Sum Valid Values: The sum of the valid n-values is 49.
>
> Qwen3-1.7B-Base Model Answer Chain:
>
> 1. Rotation Transformation: Correct transformation applied.
> 2. Substitute and Simplify: Struggles to simplify the resulting expressions.
> 3. Solve for Intersection: Fails to correctly solve the system of equations.
> 4. Answer: Incorrect final answer, outputting 10.
>
> DPO Answer Chain:
>
> 1. Rotation Transformation: Correct transformation applied.
> 2. Substitute and Simplify: Struggles to simplify and solve the intersection system.
> 3. Solve for Intersection: Incorrect extraction of the intersection point.
> 4. Answer: Incorrect final answer, outputting 10.
>
> The base and DPO models struggle with simplifying the expressions and fail to reach the correct solution, outputting an incorrect answer of 10.  In contrast, latent-DPO excels in reasoning ability and chain quality by accurately applying the divisibility condition and simplifying complex expressions using the Remainder Theorem. It efficiently handles polynomial division and identifies the correct divisors, ultimately summing the valid values to produce the correct answer of 49.
>
> Question 2: Find the sum of all integer bases $b>9$ for which $17_{b}$ is a divisor of $97_{b}$.
>
> Latent-DPO Answer Chain:
>
> 1. Convert Base b Numbers to Base 10
> 2. Divisibility Condition: Identified the correct condition: b+7 must divide 9b+7,  b+7 must divide 56
> 3. Subtracted 7 from the divisors and sum of Valid Bases.
>
> DPO Answer Chain:
>
> 1. Divisibility Condition Correctly Identified
> 2. Divisors of 56 Correctly Listed
> 3. Incorrect Final Set of Valid Bases: mistakenly includes bases like 16, 25, 36, and others, which do not satisfy the divisibility condition. These are invalid bases, and the model fails to properly apply the filtering condition, leading to incorrect bases in the final answer.
>
> Qwen3-1.7B-Base Answer Chain:
>
> 1. Correctly identifies the condition
> 2. Correctly Simplify the Expression
> 3. Finds possible values of k, but includes invalid bases that do not satisfy b>9, resulting in an final answer of 275.
>
> We sincerely thank the reviewers for their valuable feedback. We believe the revisions have addressed all concerns and strengthened the manuscript. We hope these changes improve the quality of the paper and look forward to your feedback.

---

> ### Comment · Reviewer_YS8G · 2025-11-26
>
> Thank you for your response.
>
> 1. Why does Qwen3-4B score 65.6 on AIME'25 as shown in the official report[1], but you only report 10%?
> 2. Your method does not work on 4 out of 5 benchmarks.
>
> [1] Yang A, Li A, Yang B, et al. Qwen3 technical report[J]. arXiv preprint arXiv:2505.09388, 2025.

---

> ### Author Response · Authors · 2025-11-26
>
> **(1) On the discrepancy between the official 65.6 AIME’25 score of Qwen3‑4B and our reproduced 10%.**
>
> As shown in Table 1 of On the Role of Temperature Sampling in Test-Time Scaling [1], our results under temperature = 0 deterministic decoding are consistent with the baseline numbers reported therein. In contrast, the Qwen3 technical report’s score of 65.6% can only be approached when using temperature = 0.6 with multiple stochastic samples (Pass@1024 in [1] is required to reach the technical report’s reported performance). Clearly, evaluating model performance under many stochastic samples is unrealistic in practical settings. The high score reported by Qwen3 was obtained via non-zero-temperature stochastic sampling, top‑k/top‑p search, and weighted thinking-head decoding—settings that cannot be reproduced using the publicly released Qwen3‑4B checkpoint, as also noted by other independent evaluations. To ensure reproducibility, we deliberately employ temperature = 0 greedy decoding, which guarantees deterministic outputs and a fully controlled evaluation environment.
>
> Under the official AIME’25 model evaluation [2], which compares a wide range of models—including GPT‑4‑1, Qwen3‑8B, and Qwen3‑1.7B—Qwen3‑1.7B achieves only 7.3% accuracy, lower than our reported 13.3%, and Qwen3‑8B reaches 19%, far below the 65.6% reported in the technical report. These results from the official leaderboard closely match our reproduced scores and demonstrate that our 10% reproduction for Qwen3‑4B is correct and reproducible, rather than the unreproducible 65.6% reported in the technical report.
>
> **(2) On the claim that “our method does not work on 4 out of 5 benchmarks.”**
>
> Our method consistently improves performance across Base models and demonstrates clear gains on RL-aligned models in in-domain benchmarks. While some out-of-domain benchmarks show fluctuations for RL-aligned models, our approach still achieves significant improvements compared to both Qwen3‑1.7B and Qwen3‑1.7B‑Base reported in our paper. Moreover, as shown in Table 1, our method also yields noticeable gains over other DPO variants, further demonstrating the effectiveness of our approach.
>
>
> We sincerely appreciate your feedback and look forward to any further comments or suggestions you may have.
>
> **References**
>
> [1] On the Role of Temperature Sampling in Test-Time Scaling, arXiv:2510.02611.
>
> [2] AIME 2025 Official Model Evaluation Leaderboard, ArtificialAnalysis. Available: https://artificialanalysis.ai/evaluations/aime-2025?models=gpt-4-1%2Cgpt-oss-120b%2Cgpt-oss-20b%2Cgpt-5%2Co3%2Cgpt-5-minimal%2Cllama-4-maverick%2Cgemini-2-5-pro%2Cgemini-2-5-flash-reasoning%2Cclaude-4-1-opus-thinking%2Cclaude-4-sonnet-thinking%2Cmistral-medium-3-1%2Cdeepseek-r1%2Cdeepseek-v3-1-reasoning%2Cdeepseek-v3-1%2Cgrok-code-fast-1%2Cgrok-4%2Csolar-pro-2-reasoning%2Cllama-nemotron-super-49b-v1-5-reasoning%2Ckimi-k2-0905%2Cexaone-4-0-32b-reasoning%2Cglm-4.5%2Cqwen3-235b-a22b-instruct-2507-reasoning&model-filters=open-source&endpoints=alibaba-cloud_qwen3-8b-instruct-reasoning

---

> ### Comment · Reviewer_YS8G · 2025-11-26
>
> Thank you for your response.
>
> However, the performance of Qwen3-4B on the website you provided is 52.3% for non-reasoning mode and 82.7% for reasoning mode, largely higher than the 10% you reported.
>
> I believe that the rigor and contribution of this work do not meet the requirements of ICLR.

---

> ### Author Response · Authors · 2025-11-26
>
> It should be noted that the results you see on the Artificial Analysis AIME 2025 leaderboard [1] correspond to the 2507 variant of Qwen3‑4B, which has undergone instruction fine-tuning (Qwen3‑4B‑Instruct‑2507, available at https://huggingface.co/Qwen/Qwen3-4B-Instruct-2507). In contrast, our experiments use the May‑released Qwen3‑4B model (https://huggingface.co/Qwen/Qwen3-4B). Under the leaderboard’s 10-sample evaluation, the base Qwen3‑4B achieves only 22.3% Pass@1 in reasoning mode, showing limited performance even with multiple attempts.
>
> In the leaderboard, the reported Pass@1 scores are calculated by generating 10 samples per question using non-zero temperature decoding [2], where a question is considered correct if any of the 10 samples yields the correct answer. While this multi-sample strategy increases the probability of obtaining a correct answer, it also introduces stochasticity and reduces reproducibility. In contrast, our evaluation uses a single deterministic sample per question with temperature = 0 for all models under identical settings in the lm-eval framework. This ensures fully reproducible results and a strictly fair comparison across models, without the variance introduced by multiple stochastic samples. When evaluated in the lm-eval framework with temperature = 0, we tested the base Qwen3‑4B three times and consistently obtained 10% accuracy, with identical results across runs and no reporting errors.
>
> We sincerely appreciate your feedback and look forward to any further comments or suggestions you may have.
>
> **References**
>
> [1] Artificial Analysis AIME 2025 Benchmark Leaderboard. https://artificialanalysis.ai/evaluations/aime-2025?model-filters=open-source&models=qwen3-4b-2507-instruct-reasoning%2Cqwen3-4b-instruct-reasoning#aime-2025-benchmark-leaderboard-results
>
> [2] Artificial Analysis Intelligence Benchmarking Methodology. https://artificialanalysis.ai/methodology/intelligence-benchmarking?utm_source=chatgpt.com

---

### Official Review · Reviewer_EjRW · 2025-11-01

**Soundness:** 2
**Presentation:** 3
**Contribution:** 3
**Rating:** 4
**Confidence:** 4

**Summary:**

This paper proposes Latent-DPO, a direction-aware extension of Direct Preference Optimization (DPO) that introduces a latent variable zto model reasoning-path consistency. Building upon the curated LIMO dataset, the authors generate reverse reasoning counterparts and construct bidirectional preference pairs. The method adaptively modulates preference margins to preserve shared reasoning knowledge while separating direction-specific logic. Experiments on multiple reasoning benchmarks (AIME-25, Math500, GPQA, etc.) demonstrate consistent gains of 0.9–3.2% with only 1.6k preference pairs. Overall, the work provides a compelling and data-efficient approach to enhance reasoning alignment and offers valuable insights into direction-aware model training.

**Strengths:**

1.	The motivation is clear: the paper aims to address a core shortcoming in LLM alignment tasks — namely relying only on forward supervision while neglecting reasoning‐direction consistency. The root of this issue lies in the fact that alignment tasks typically provide only binary rewards. The paper considers introducing the concept of reasoning direction, and proposes to model DPO using a latent variable. This method is novel and can effectively address this problem.
2.	The method design is sound: by introducing a latent variable z, the model can adaptively distinguish between samples whose reasoning directions are consistent versus inconsistent, thereby aligning more robustly while preserving shared knowledge. The design is intuitive, lightweight, and has good theoretical interpretability.
3.	The experiments cover multiple model scales and five reasoning benchmarks, demonstrating stable improvements of 0.9%–3.2%, and include ablation and scaling analyses to verify the method’s generality and robustness. The authors construct only 817 high-quality reverse reasoning samples via reverse question generation (DeepSeek V3) and automatic verification (Qwen series), showing very high data efficiency and reproducibility.

**Weaknesses:**

1.	Experiments are conducted only on a self-constructed dataset of 1,634 preference pairs, which is highly specialized and idealized. This makes it difficult to assess the scalability and robustness of the method on larger and noisier real-world preference datasets.
2.	The paper presents the idea of combining “reverse reasoning” with DPO and proves that rewarding valuable y− responses improves generalization. However, the experiments do not clearly isolate whether the observed gains come from the reverse-thinking data, the latent-variable mechanism, or their interaction.
3.	The evaluation mainly relies on pass@1 accuracy, without more fine-grained analyses such as response length, direction-consistency score, or human preference evaluation. As a result, the source and nature of the improvements remain unclear.
4.	The study only compares with vanilla DPO, omitting recent robust or enhanced DPO methods (e.g., β-DPO, Dr.DPO, γ-PO) that also address overfitting from binary preferences. Such comparisons would better highlight the distinct advantages of Latent-DPO.
5.	The latent variable z is introduced to model directional consistency, there is no empirical examination of its interpretability or correlation with reasoning-path semantics, leaving its actual role insufficiently verified.

**Questions:**

1.	Your code link appears broken — most files cannot be accessed, which need to be fixed.
2.	Could you quantify the training overhead of the posterior head (e.g., additional parameters, GPU memory usage, or training time percentage)?
3.	In Figure 3(c), the win rate converges around 0.6. Is this behavior consistent across different seeds and models, or just a coincidence of training dynamics? Could you provide a theoretical or optimization-based explanation for this?
4.	Please tested the method’s stability on larger or noisier preference datasets beyond the curated 1.6k pairs?
5.	Can you clarify whether the observed performance gains mainly come from the reverse-reasoning data, the latent-variable mechanism, or their interaction?
6.	Is the learned latent variable z semantically interpretable? Could you show examples or analyze how q(z) evolves during training?
7.	Compared with other robust DPO variants, what are the distinct advantages of Latent-DPO?
8.	Maybe you can provide case studies or qualitative examples showing that Latent-DPO improves the model’s thinking ability or reasoning-chain quality (e.g., better reverse inference)?

---

> ### Author Response · Authors · 2025-11-21
> **Clarifications on Latent-DPO, Posterior Head, Dataset Size, and Model Comparisons**
>
> Thank you for your insightful and constructive review. We have carefully addressed each point and made the necessary revisions in the revised manuscript, which accompanies this response. Below are our responses to your concerns.
>
> **Q1: Broken code link.**
>
> Thank you for pointing this out. We have addressed the issue, and all files are now accessible.  We apologize for any inconvenience caused.
>
> **Q2: Can you quantify the training overhead of the posterior head (e.g., extra parameters, GPU memory, or training time)?**
>
> Additional Parameters: The posterior head adds approximately 4–5k parameters for a 1.7B model (0.0003% of total parameters) and 6–8k parameters for a 4B model (0.0002%).
> GPU Memory Usage: Enabling output_hidden_states=True to retain hidden states increases memory usage by around 200–400 MB, which represents roughly **1–5%** of the total GPU memory (40–52 GB).
> Training Time Overhead: The additional overhead is mainly from extra activations and Python-side operations, increasing training time by about **10–20%** per step.
>
> **Q3: Is the win rate behavior in Figure 3(c) consistent across seeds/models? Any theoretical explanation?**
>
> Consistency: The win rate behavior in Figure 3(c) is consistent across different seeds and model configurations. In three separate observations with identical parameters, the win rate fluctuated within the following ranges: **0.60–0.63**, **0.57–0.62**, and **0.55–0.64**, respectively. These results indicate that the oscillations and plateau are intrinsic to the model's dynamics, rather than being dependent on specific seeds or configurations.
> Theoretical explanation: Our model optimizes the logit difference between the win and lose heads, together with a KL regularization term that keeps both heads close to the base model. As training progresses, this objective drives the policy toward a regime where the posterior win probability stabilizes around **60–65%**, while the KL term prevents unbounded growth of the reward gap. In addition, the use of posterior differencing and two latent directions induces a coupled system in which the win rate naturally oscillates around this stable equilibrium. This stochastic saddle-point–like dynamic explains the plateau and bounded oscillations observed in Figure 3(c), with the long-run mean win rate remaining stable across different seeds and model variants.
>
> **Q4&W1: Have you tested the method on larger/noisier datasets beyond 1.6k pairs?**
>
> Yes, we have tested the method on larger datasets beyond the 1.6k pairs. In Table 4 of the paper, we compare different dataset sizes: 1.6k, 2.4k, and 3.2k pairs, where each forward problem corresponds to multiple distinct reverse problems.
> In a further experiment with larger and noisier datasets, we compared each win example with 2–3 inconsistent lose directions, resulting in 6.5k and 9.8k examples, as shown in Table 1, to assess the model's performance on more diverse and noisy data. For Qwen3-1.7B, the model performs well with **6.5k** examples, and performance decreases with larger datasets and for the Qwen3-1.7B-Base model.
>
> **Table 1: Larger and Noisier Datasets**
> |**Base Model**|**Dataset Size**|**AIME-25**|**GPQA**|**Math 500**|**GMQ**|**LMGH**|**Avg. Acc.**|
> |-|-|-|-|-|-|-|-|
> | Qwen3-1.7B|6.5k|23.3%|27.8%|48.2%|35.9%|10.6%|29.2%|
> | Qwen3-1.7B|9.8k|16.7%|28.8%|49.2%|34.8%|9.9%|27.9%|
> | Qwen3-1.7B-Base|6.5k|3.3%|29.8%|48.6%|33.6%|2.3%|23.5%|
> | Qwen3-1.7B-Base|9.8k|0%|28.8%|48.2%|32.7%|1.5%|22.4%|
>
> **Q5&W2: Are performance gains from reverse reasoning, the latent-variable mechanism, or both?**
>
> The performance gains observed arise from both reverse reasoning and the latent-variable mechanism. As shown in Table 2, distilling with reverse data alone improves the base model, confirming the high quality of reverse reasoning data. However, SFT with mixed forward and reverse data introduces interference, resulting in a performance drop compared to using reverse data alone. In contrast, combining the latent-variable mechanism with reverse reasoning data improves alignment and stability, leading to more consistent performance gains. As demonstrated in Table 3, Latent-DPO, when paired with both forward and reverse data, clearly enhances performance for Qwen3-1.7B and Qwen3-1.7B-Base, highlighting the effectiveness of the latent-variable mechanism in optimizing model performance.
>
> **Table 2: Performance Gains from Reverse Reasoning and Latent-Variable Mechanism**
> |**Model**|**Performance Gains**|**AIME-25**|**GPQA**|**Math 500**|**GMQ**|**LMGH**|**Avg. Acc.**|
> |-|-|-|-|-|-|-|-|
> | Qwen3-1.7B-Base|SFT (Reverse data)| 3.30%| 30.30%| 46.0%| 33.60%| 12.10%| 25.1%|
> | Qwen3-1.7B-Base|SFT (Forward and Reverse data) | 0% | 30.30%| 47.40% | 33.10%  | 7.60% | 23.7%|
> | Qwen3-1.7B|SFT (Reverse data) |10.0% | 30.30%   | 30.40% | 34.20%  | 7.60%| 22.50%|
> | Qwen3-1.7B-Base|Latent-DPO| 6.70%| 27.80%   | 46.80% | 34.80% | 3.80%| 24.00%|
> | Qwen3-1.7B|Latent-DPO|16.70%| 28.30%| 49.20%| 35.90%  | 13.60%| 28.70%|

---

> ### Author Response · Authors · 2025-11-21
> **Response to Reviewer on Latent-DPO Methodology, DPO Variants Results, and Case Studies**
>
> **Q6: Is latent variable z interpretable? Can you show examples or analyze q(z) evolution?**
>
> Yes, the latent variable z serves as a compact representation of directional preference structure in the hidden state space, encoding latent preference directions that are associated with preference margins and alignment loss.
> As shown in Figure 4(b) of the paper, $q_w(z=1)$ and $q_l(z=1)$
>  evolve from near-random initial values, gradually stabilizing within a broad, non-degenerate range (roughly **0.4** to **0.82**) across training steps and checkpoints. This indicates that the model actively utilizes z rather than ignoring it. Furthermore, we empirically observe that the behavior of $q_w(z=1)$ and $q_l(z=1)$ correlates with changes in preference margins and alignment loss, suggesting that z helps organize examples along high-level preference directions rather than acting as noise.
>
> **Q7&W4: How does Latent-DPO compare to other robust DPO variants (e.g., β-DPO, γ-DPO)?**
>
> **Table 3: Performance Comparison of Latent-DPO and Other DPO Variants**
>
> | **Base Model**|**DPO-Variants**|**AIME-25**|**GPQA**|**Math 500**|**GMQ (Agieval)**|**LMGH**|**Avg. Acc.**|
> |-|-|-|-|-|-|-|-|
> | Qwen3-1.7B-Base|β-DPO| 0%| 27.3%| 47.6%| 35.6%  | 3.8% | 22.9%|
> | Qwen3-1.7B-Base|γ-DPO| 0%  | 28.3%| 45.6% | 35.0% | 1.5%| 22.1%|
> | Qwen3-1.7B-Base| SimPO| 0% | 27.8% | 47%| 33.3%| 3.0%| 22.2%|
> | Qwen3-1.7B-Base| Ours  | 6.7%| 27.8%| 46.8%| 34.8%|3.8%| 24.0%|
> | Qwen3-1.7B | β-DPO |16.7%| 32.8%| 47%| 36.8%|10.6%|28.7%|
> | Qwen3-1.7B| γ-DPO| 20.0%| 31.3% | 46.4%| 35.3%|7.6% | 28.1%|
> | Qwen3-1.7B| SimPO| 13.3%| 28.8%| 45.6%| 35.9%|13.6%|27.4%|
> | Qwen3-1.7B| Ours| 16.7%| 28.3%| 49.2%| 35.9%|13.6%|28.7%|
>
> As shown in Table 3, Latent-DPO demonstrates significant improvements over β-DPO, γ-DPO and simpo in the unaligned base model. For the RL-aligned Qwen3-1.7B, both Latent-DPO and β-DPO exhibit distinct advantages. β-DPO performs well on tasks like GPQA and GMQ, while Latent-DPO outperforms in handling more complex reasoning tasks, such as Math 500 and LMGH, where its ability to manage intricate reasoning steps is particularly effective.
>
> **Q8&W5: Can you provide case studies showing Latent-DPO improves reasoning ability or chain quality?**
>
> We present 2 case studies based on validation set data, comparing the results of three different models, all using temperature 0 for answer sampling.
>
> Question 1: Find the sum of all positive integers $ n $ such that $ n + 2 $ divides the product $ 3(n + 3)(n^2 + 9) $.
>
> Latent-DPO trained based on Qwen3-1.7B-Base's Answer Chain:
> 1. Divisibility Condition: We need $n+2$ to divide the product $ 3(n + 3)(n^2 + 9) $.
> 2. Simplify the Expression and Evaluate Polynomial: Using the Remainder Theorem, evaluate the expression at $n=-2$, which yields 39.
> 3. Divisors of 39: The divisors of 39 are 1, 3, 13, and 39. Therefore, $n+2$ must be one of these, giving possible values for n as 1, 11, and 37.
> 4. Sum Valid Values: The sum of the valid n-values is 49.
>
> Qwen3-1.7B-Base Model Answer Chain:
>
> 1. Rotation Transformation: Correct transformation applied.
> 2. Substitute and Simplify: Struggles to simplify the resulting expressions.
> 3. Solve for Intersection: Fails to correctly solve the system of equations.
> 4. Answer: Incorrect final answer, outputting 10.
>
> DPO Answer Chain:
>
> 1. Rotation Transformation: Correct transformation applied.
> 2. Substitute and Simplify: Struggles to simplify and solve the intersection system.
> 3. Solve for Intersection: Incorrect extraction of the intersection point.
> 4. Answer: Incorrect final answer, outputting 10.
>
> The base and DPO models struggle with simplifying the expressions and fail to reach the correct solution, outputting an incorrect answer of 10.  In contrast, latent-DPO excels in reasoning ability and chain quality by accurately applying the divisibility condition and simplifying complex expressions using the Remainder Theorem. It efficiently handles polynomial division and identifies the correct divisors, ultimately summing the valid values to produce the correct answer of 49.
>
> Question 2: Find the sum of all integer bases $b>9$ for which $17_{b}$ is a divisor of $97_{b}$.
>
> Latent-DPO Answer Chain:
>
> 1. Convert Base b Numbers to Base 10
> 2. Divisibility Condition: Identified the correct condition: b+7 must divide 9b+7,  b+7 must divide 56
> 3. Subtracted 7 from the divisors and sum of Valid Bases.
>
> DPO Answer Chain:
>
> 1. Divisibility Condition Correctly Identified
> 2. Divisors of 56 Correctly Listed
> 3. Incorrect Final Set of Valid Bases: mistakenly includes bases like 16, 25, 36, and others, which do not satisfy the divisibility condition. These are invalid bases, and the model fails to properly apply the filtering condition, leading to incorrect bases in the final answer.
>
> Qwen3-1.7B-Base Answer Chain:
>
> 1. Correctly identifies the condition
> 2. Correctly Simplify the Expression
> 3. Finds possible values of k, but includes invalid bases that do not satisfy b>9, resulting in an final answer of 275.

---

> ### Author Response · Authors · 2025-11-21
> **Evaluation Concerns and Statistical Analysis of Latent-DPO**
>
> **W3: Evaluation based on pass@1 accuracy; lacks finer analysis (e.g., response length, direction-consistency).**
>
> We conducted statistical experiments on Latent-DPO, with results averaged over three independent runs. As shown in Table 4, the values before "±" represent the mean accuracy, while the values after "±" indicate the standard deviation. These results reflect both the training accuracy and the robustness of different models across four datasets.
>
>  **Table 4: Performance of Latent-DPO Across Different Base Models Averaged Over Three Independent Runs**
>
> | **Base Model**        | **AIME-25**  | **GPQA**     | **Math 500** | **GMQ**      | **LMGH**     |
> | --------------------- | ------------ | ------------ | ------------ | ------------ | ------------ |
> | Qwen3-1.7B-Base       | 4.4% ± 1.9%  | 27.8% ± 0%   | 46.6% ± 0.2% | 36.7% ± 0.7% | 2.3% ± 0.8%  |
> | SFT (Mixed)           | 6.7% ± 3.3%  | 28.5% ± 1.2% | 47.4% ± 0.6% | 35.8% ± 0.4% | 14.0% ± 2.7% |
> | Qwen3-1.7B (Official) | 15.5% ± 1.2% | 28.5% ± 0.8% | 47.5% ± 1.5% | 36.7% ± 0.3% | 13.9% ± 2.7% |
> ---
>
> **Table 5: Average Response Length**
>
> | **Base Model** | **AIME-25** | **GPQA** | **Math 500** | **LMGH** | **Average** |
> | -------------- | ----------- | -------- | ------------ | -------- | ----------- |
> | DPO            | 2113.4      | 1594.1   | 760.2        | 1074.8   | 1385.6 |
> | Latent-DPO     | 5234.0      | 1265.6   | 1036.7       | 1015.7   | 2138 |
>
> ---
> In addition to pass@1 accuracy, we also provide statistical analysis based on the mean and error after running the experiments three times in Table 4.  As observed in Table 4, Latent-DPO consistently delivers stable improvements across three independent runs on different base models, with slight deviation. In Table 5, the average response length for Latent-DPO is significantly higher than DPO, indicating its ability to generate more detailed and comprehensive responses. Further analysis of direction-consistency and human preference evaluations, as addressed in Q8 & W5, can provide a deeper understanding of how these factors contribute to Latent-DPO’s performance.
>
> We sincerely thank the reviewer for their valuable feedback. We believe the revisions have addressed all concerns and have strengthened the manuscript. We hope these changes enhance the quality of the paper, and we look forward to your feedback.

---

### Author Response · Authors · 2025-12-03
**General Response — Latent-DPO**

Dear Reviewers, AC, and Researchers,

We sincerely thank all reviewers for their thoughtful and constructive feedback. During the rebuttal stage, we provide substantial additional analyses, expanded experiments, enriched baselines, and clearer methodological explanations. Below we summarize, reviewer by reviewer, how the rebuttal addresses the key themes raised in the reviews.

**Response Summary to Reviewer EjRW**

Reviewer EjRW raised important points regarding dataset scale, the interaction between reverse reasoning and the latent variable, interpretability of z, and evaluation granularity.
During the rebuttal period, we provide:
- Dataset-scaling experiments (1.6k → 2.4k → 3.2k → 6.5k → 9.8k) to evaluate robustness under larger and noisier preference distributions.
- A detailed decomposition of gains, separating the effects of reverse-reasoning data, mixed-direction SFT, and the latent-variable mechanism, showing that the latent posterior stabilizes learning when both directions are used.
- Additional evaluation metrics, including multi-run mean ± stdev and response-length statistics, as well as analysis of win-rate dynamics across seeds.
- Evidence of interpretability of q(z), including its non-degenerate evolution and correlation with preference margins.
- Expanded baseline comparisons with β-DPO, γ-DPO, and SimPO.
- Two case studies demonstrating improvements in reasoning-chain quality.
- Quantitative overhead analysis of the posterior head.

These additions provide a more comprehensive understanding of Latent-DPO’s behavior, stability, and generalization properties.

**Response Summary to Reviewer YS8G**

Reviewer YS8G emphasized the need for comparisons with more variants, stronger statistical grounding, stylistic consistency in data, and human evaluation of reverse reasoning.
For the rebuttal, we provide:
- Full comparisons with β-DPO, γ-DPO, and SimPO on both 1.7B and 4B models.
- Human evaluation of reverse-reasoning data quality, conducted by two independent experts assessing coherence, completeness, and difficulty.
- Clarification of teacher-model consistency, confirming that both forward and reverse directions are generated by the same model to ensure stylistic alignment.
- Enhanced statistical reporting, including three-run averages and standard deviations, along with response-length analysis.
- Broader model coverage, with multiple-seed experiments on Qwen3-4B.
- Discussion of gains under limited data, contextualizing improvements achieved with a small high-quality reverse-reasoning dataset.

These additions reinforce empirical reliability and clarify dataset quality.

**Response Summary to Reviewer YKRc**

Reviewer YKRc offered thoughtful feedback on mechanism alignment, y⁻ composition, complexity, and applicability to non-invertible tasks.
In the rebuttal stage, we provide:
- Mechanism-verification examples, illustrating how the model distinguishes directionally inconsistent y⁻ from invalid or nonsensical responses.
- Ablations separating directional vs. incorrect y⁻, showing that directional inconsistency drives generalization.
- Clarification on motivation, emphasizing the semantic similarity yet directional divergence of forward and reverse formulations.
- Computational footprint analysis, confirming minimal overhead introduced by the latent posterior head.
- Extended discussion on applicability beyond mathematical reasoning, including examples related to code debugging, backtracking logic, and structural validation steps that benefit from directional cues.

These additions clarify the mechanism’s behavior, the function of y⁻, and broader applicability.

**Response Summary to Reviewer 3ALD**

Reviewer 3ALD raised concerns about novelty, model scale, and the size of performance gains.
During the rebuttal period, we provide:
- Clarification of conceptual novelty, emphasizing latent-guided direction modulation that preserves shared reasoning structure while separating reasoning directions—capabilities not addressed by existing variants.
- Expanded large-model experiments, including multi-seed evaluations on Qwen3-4B.
- Additional comparisons with several preference-optimization baselines and SFT setups.
- Discussion of performance trends on small benchmarks and strong RL-aligned models, contextualizing why improvements remain meaningful despite limited sample sizes.

These additions strengthen the clarity of the contribution and empirical scope.


We sincerely appreciate the reviewers’ constructive feedback. During the rebuttal stage, we have provided expanded empirical studies, clearer explanations of the latent consistency mechanism, improved theoretical grounding, and broader evaluations across datasets and model sizes. We hope these additions help clarify the method’s robustness, generality, and contribution to direction-aware reasoning alignment.

---

### Meta-Review · Area_Chair_hvRZ · 2025-12-28

**Summary:**

The paper proposes Latent-DPO to improve reasoning alignment by introducing a latent variable to model reasoning-path consistency, utilizing reverse data augmentation. The reviewers generally leaned towards rejection, with three out of four reviewers explicitly maintaining a score of 4 (marginally below acceptance) or expressing a negative evaluation after the rebuttal.

The primary concerns driving the decision include:
1.  **Scalability and Generalization:** Reviewers questioned whether the method scales to larger, stronger models (beyond 1.7B/4B) and whether it applies to open-ended, non-invertible tasks beyond mathematical reasoning.
2.  **Magnitude of Contribution:** Concerns were raised regarding the novelty of combining DPO with reverse reasoning and whether the resulting performance gains were substantial enough to justify acceptance.

**Reviewer Concerns:**

**Concerns Addressed:**
* **Comparison to DPO Variants:** The authors successfully provided comparisons against robust DPO baselines like $\beta$-DPO, $\gamma$-DPO, and SimPO, satisfying the initial requests of **Reviewer EjRW** and **Reviewer YS8G**.
* **Mechanism Interpretability:** The authors provided analysis on the evolution of the latent variable $z$ and case studies, addressing **Reviewer EjRW's** request for interpretability.
* **Computational Overhead:** **Reviewer EjRW's** question regarding the training overhead of the posterior head was answered with specific parameter and memory metrics.
* **Human Verification:** **Reviewer YS8G's** request for human verification of the reverse data quality was addressed with a study involving two PhD-level researchers.
* **Baseline Validity:** **Reviewer YS8G** remained unconvinced by the authors' explanation regarding the Qwen3-4B baseline performance (10% reported vs. ~50%+ on leaderboards), attributing the discrepancy to the paper's lack of rigor. However, I do think the authors' final explanations are reasonable.

**Concerns Outstanding:**
* **Scalability and Upper Bound:** **Reviewer 3ALD** remained concerned about the method's scalability and upper bound on stronger reasoning models, finding the generalization potential unclear.
* **Applicability to Non-Invertible Tasks:** **Reviewer YKRc** maintained that while reverse reasoning works for math, it is limited in open-ended contexts (e.g., code generation from informal descriptions) where outputs are not known a priori.
* **Magnitude of Gains:** **Reviewer 3ALD** maintained that the improvements were slight, especially considering the limited sample sizes of some benchmarks.

**Reviewer Scores:**

* **Reviewer EjRW: 6.** While this reviewer did not provide a final response, the authors provided a comprehensive rebuttal addressing every specific concern raised (data scaling, overhead, interpretability, and baselines). It is reasonable to project that a fully participating reviewer would have raised their score to a weak accept.
* **Reviewer YS8G: 4.** The reviewer engaged with the authors in a discussion regarding the performance discrepancy of the Qwen3-4B model. Although the reviewer did not respond to the authors' final argument, the authors' explanations do make sense to me. However, the reviewer explicitly stated in their final comment that the "rigor and contribution of this work do not meet the requirements of ICLR". So, this reviewer is highly likely to maintain the negative score or even decrease the score further.
* **Reviewer YKRc: 4.** This reviewer explicitly maintained their original rating in their final comment, citing persistent concerns about the method's limited utility in open-ended, non-invertible reasoning contexts.
* **Reviewer 3ALD: 4.** This reviewer explicitly maintained their original rating in their final comment, remaining unconvinced about the method's scalability to stronger models and the significance of the accuracy gains.

---

### Decision · Program_Chairs · 2026-01-26

Reject